# Research Progress of Metal Anticancer Drugs

**DOI:** 10.3390/pharmaceutics15122750

**Published:** 2023-12-11

**Authors:** Yun Bai, Gerile Aodeng, Lu Ga, Wenfeng Hai, Jun Ai

**Affiliations:** 1Inner Mongolia Key Laboratory of Environmental Chemistry, College of Chemistry and Enviromental Science, Inner Mongolia Normal University, 81 Zhaowudalu, Hohhot 010022, China; 15024820597@163.com (Y.B.); aodeng@imnu.edu.cn (G.A.); 2College of Pharmacy, Inner Mongolia Medical University, Jinchuankaifaqu, Hohhot 010110, China; 13404832082@163.com; 3Inner Mongolia Key Laboratory of Carbon Nanomaterials, Nano Innovation Institute (NII), College of Chemistry and Materials Science, Inner Mongolia Minzu University, Tongliao 028000, China

**Keywords:** platinum anticancer drugs, ruthenium anticancer drugs, iridium anticancer drugs, gold anticancer drugs, cancer

## Abstract

Cancer treatments, including traditional chemotherapy, have failed to cure human malignancies. The main reasons for the failure of these treatments are the inevitable drug resistance and serious side effects. In clinical treatment, only 5 percent of the 50 percent of cancer patients who are able to receive conventional chemotherapy survive. Because of these factors, being able to develop a drug and treatment that can target only cancer cells without affecting normal cells remains a big challenge. Since the special properties of cisplatin in the treatment of malignant tumors were accidentally discovered in the last century, metal anticancer drugs have become a research hotspot. Metal anticancer drugs have unique pharmaceutical properties, such as ruthenium metal drugs with their high selectivity, low toxicity, easy absorption by tumor tissue, excretion, and so on. In recent years, efficient and low-toxicity metal antitumor complexes have been synthesized. In this paper, the scientific literature on platinum (Pt), ruthenium (Ru), iridium (Ir), gold (Au), and other anticancer complexes was reviewed by referring to a large amount of relevant literature at home and abroad.

## 1. Introduction

Cancer is a major threat to human health. It is a serious disease that has a great impact on human health [1]. In some developed countries, the number of deaths from cancer is increasing [2]. According to the World Health Organization (WHO), oncological diseases have become the second-leading cause of death worldwide. Global cancer deaths in 2020 will reach 9.96 million, of which 5.53 million will be men and 4.43 million will be women. The number of new cases of breast cancer in the world has reached 2.26 million, exceeding the number of cases of lung cancer. Therefore, breast cancer has replaced lung cancer as the most common type of cancer in the world. Therefore, to cure cancer is our common hope, and for this goal, research on anticancer drugs is essential. At present, the methods of treating cancer include targeted anticancer drug therapy, chemotherapy, hormone treatment, radiotherapy, surgery, and so on. These treatments are selected and combined according to the patient’s specific condition and tumor type to achieve the best treatment results [3,4]. Nanotechnology has enormous potential in the prognosis, diagnosis, and drug delivery of cancer, and is therefore considered to have significant applications in clinical treatment [5]. New photodynamic therapy is a treatment process that involves injecting a patient with a light-sensitive drug and then irradiating it using the maximum absorption wavelength to generate locally reactive oxygen species (ROS) such as singlet oxygen [6]. The aim is to induce significant toxicological effects at the site of the tumor tissue, leading to necrosis and ultimately cell death [7]. Transition metals play a very important role in biochemistry as cofactors in the active sites of enzymes, enabling a large number of selective catalytic transitions required to sustain biological processes [8]. Transition metals naturally occur in trace amounts within living organisms. Nevertheless, overconsumption of these elements can lead to numerous detrimental effects, including cancer [9]. However, it is this toxicological potential that lays the foundation for transition-metal-based anticancer therapies [4]. Besides their extensively researched cytotoxic and anticancer properties, metal-based drugs are commonly employed in cancer-immune interactions and have the ability to reverse pivotal aspects of immune evasion [10]. At present, there are dozens of drugs for the clinical treatment of malignant tumors, but affected by the factors of cancer pathogenesis, the cure rate of cancer is low, and the therapeutic drugs are rare. Therefore, the study of anticancer drugs and related mechanisms of action has high clinical significance. Platinum-based drugs, particularly cisplatin, a first-generation platinum-based anticancer drug, are the most widely used anticancer drugs in clinical practice, with established efficacy in head, neck, and reproductive cancers [11]. Carboplatin, the second generation of platinum anticancer drug after cisplatin, was launched in 1989. The anti-tumor spectrum of this drug is relatively close to that of cisplatin, and the adverse reactions are mild. In 2002, oxaliplatin, the third generation of platinum drugs, was used in the treatment of colon cancer, and then new platinum drugs such as nedaplatin and lobaplatin were successively used in clinical treatment. For example, BBR-3464 can be used for the treatment of advanced cancer patients resistant to some common platinum anticancer drugs. The anticancer mechanism is shown in Figure 1; it has a higher degree of cross-linking with DNA compared with cisplatin, as well as powerful anticancer efficacy and no cross-resistance. However, in the development process of such metal anticancer drugs, difficulties are also encountered, such as hepatotoxicity, neurotoxicity, etc., which seriously restrict the efficacy of drug use [12]. Since cisplatin’s anti-proliferative properties were discovered more than half a century ago, intracellular release of poisonous metal ions such as platinum(II)/(IV), Au(I)/(III), and Ru(II)/(III) has become a major component of metal-based anticancer therapy [13]. Although there are many chemotherapy methods and adjuvant anti-tumor drugs used in clinical research, some of them have significant effects on the treatment of cancer, but many drugs can only alleviate the pain of patients, while being unable to cure cancer. Therefore, domestic researchers have spent a large amount of resources on the research and development of anti-tumor drugs, hoping to have major breakthroughs in the future and striving for a radical cure for cancer. In this paper, the scientific literature on anticancer complexes such as platinum, ruthenium, iridium, and gold were reviewed.

## 2. Metal Anticancer Drugs

### 2.1. Platinum Anticancer Drugs

In the 1960s, the American scientist Rosenberg observed for the first time that platinum compounds could inhibit cell growth in an experiment studying the effect of electric fields on bacterial growth, thus opening the prelude to the development of this unique configuration of anti-tumor drugs. Figure 2 shows the development history of classical platinum anticancer drugs. As an important class of metal chemotherapy drugs, platinum anticancer drugs are used in more than half of chemotherapy. Currently, platinum anticancer drugs that have been approved globally for market include cisplatin, carboplatin, oxaliplatin, multi-nuclear platinum, with nedaplatin, lobaplatin, and heptaplatin approved in some countries.

#### 2.1.1. The First Class of Platinum Drugs

Cisplatin, known chemically as cis-dichlorodiammine platinum (II) (shown in Figure 3), belongs to the first generation of platinum-based metal anticancer drugs. Cisplatin, which was approved by the U.S. Food and Drug Administration in 1978, has strong anticancer activity [15], so it can be used in the treatment of different malignant tumors, mainly for lung cancer, esophageal cancer, breast cancer, gastric cancer, malignant lymphoma, and ovarian cancer, and head and neck tumors [15]. Platinum-based anticancer drugs are cell-cycle non-specific drugs, and DNA is identified as the main cell target. Attracted by the static electricity of DNA, cisplatin hydrate attacks the bases on the DNA chain, especially the 7 N atoms of guanine and adenine, forming cross-linked products in the chain, thus changing the function of the normal replication template of DNA and causing DNA replication dysfunction(as shown in Table 1). Therefore, it affects the division of cancer cells (the anticancer mechanism is shown in Figure 4) [16]. At the same time, it can also cause ROS to accumulate in mitochondria and activate the mitochondria-dependent apathetic pathway to induce cell death [17]. Oxidative stress is one of the main mechanisms by which cisplatin exerts anti-tumor effects. Cisplatin can not only bind and consume reducing substances in cells, but also form complexes with mitochondrial DNA, resulting in mitochondrial dysfunction and increased ROS production, thus inducing oxidative stress and ultimately cell death [18].

However, in the process of tumors receiving continuous cisplatin stimulation and eventually evolving into drug-resistant cells, ROS metabolism abnormalities are closely related to the occurrence of cisplatin resistance. On the one hand, cisplatin-resistant cells reduce cisplatin-induced oxidative stress by promoting cisplatin efflux, reducing ROS production and increasing the synthesis of reducing substances such as glutathione, and enhancing their resistance to cisplatin. On the other hand, intracellular ROS is also involved in the regulation of multiple links of cisplatin action, including cisplatin transport, DNA damage repair, and signaling pathways. These adverse reactions have led to great limitations in the cisplatin treatment of malignant tumors. At the same time, cisplatin has strong side effects, including gastrointestinal reactions (mainly severe nausea and vomiting), renal toxicity (mainly damage to renal tubules), neurotoxicity (mainly muscle pain and movement disorders), ototoxicity (mainly tinnitus and hearing loss), bone marrow suppression and allergic reactions, etc. There is also resistance to cisplatin in some tumors [19,20]. Therefore, to improve resistance and reduce these adverse reactions is the direction of further research on platinum drugs [21].

In recent years, cisplatin has often been used in combination with other drugs to treat breast cancer, so as to reduce cisplatin resistance or alleviate adverse reactions, and improve its clinical efficacy. Since natural active ingredients can significantly inhibit the recurrence and metastasis of breast cancer, reverse drug resistance, regulate the immune function of the body, reduce the toxicity of cisplatin, improve the quality of life of patients, and prolong the survival period, their application in the treatment of breast cancer is increasing [22,23]. For example, natural flavonoids have anti-tumor, anti-inflammatory, antioxidant, and other pharmacological activities, and are ideal drugs for tumor prevention or clinical multi-drug combination therapy. Therefore, flavonoids combined with cisplatin can improve the anti-breast cancer effect of cisplatin. A variety of natural alkaloids also have anti-tumor, antibacterial, and other biological activities, and can reduce the adverse reactions of chemotherapy drugs and improve the body’s immune function [24]. Polyphenols are also effective natural anti-tumor protective and therapeutic agents, which can directly kill tumor cells, and can also play a synergistic anti-breast cancer role with cisplatin by inhibiting angiogenesis, inducing cell cycle arrest, and affecting signal pathways to inhibit cell proliferation and migration [25]. Natural terpenoids also have strong anti-tumor activity [26] and a synergistic anti-breast cancer effect when combined with cisplatin. Other natural active ingredients such as polysaccharides have also been combined with cisplatin for anti-breast cancer research [27]. Shenqi Fuzheng injection can down-regulate the expression of P-glycoprotein in MDA-MB-231 cells resistant to cisplatin, improve cisplatin sensitivity, reduce the release of IL-10 and PGE2 induced by cisplatin, enhance immunity, and thus exert a good synergistic anti-breast cancer effect [28,29]. There are many studies on the application of cisplatin in combination with other chemotherapy agents in the treatment of breast cancer, both at the cellular level and in clinical patients. The results of the studies mainly show synergistic effects and reduction or reversal of cisplatin resistance. For example, gemcitabine combined with cisplatin activates the mTOR/S6K1/NF-κB signaling pathway, up-regulates the expression of NF-κB protein and mRNA, down-regulates mTOR and S6K1 proteins, changes energy metabolism, inhibits cell proliferation, and promotes cleavage and death of breast cancer cells [30]. For example, one study mixed Gd-Pt therapeutic agents that incorporate platinum into micelles or other types of nanoparticles [31]. Xin et al. studied the efficacy of alanine-proline-arginine-proline-glycine (APRPG) peptide-coupled polyglycol cationic liposome coated with zoledronic acid (ZOL) (APRPG-PEG-ZOL-CLPS) for vascular normalization [32]. Cisplatin is used to improve the anticancer effect. Studies have shown that APRPG-PEG-ZOL-CLPs improves anticancer effects, which is thought to normalize blood vessels. The results showed that APRPG-PEG-ZOL-CLPs could significantly inhibit the activity, migration, and tube formation of human umbilical vein endothelial cells (HUVECs). In addition, APRPG-PEG-ZOL-CLPs decreased tumor vascular density, decreased hypoxia-inducing factor 1α(HIF-1α), and increased the expression of thrombus response protein-1 (TSP-1). Therefore, the anticancer effect of APRPG-PEG-ZOL-CLPs in combination with cisplatin is better than that of PEG-ZOL-CLP or ZOL in combination with cisplatin, and the tumor volume is significantly reduced. Therefore, the APRPG-PEG-ZOL-CLP combined cisplatin regimen is the most effective in regulating the tumor vascular system and improving the efficacy of antitumor drug therapy. Using Raman spectroscopy, Jie et al. [33] studied the effects of a combination of a γ-secretase inhibitor (DAPT) and cisplatin on osteosarcoma (OS) cells (the anticancer mechanism is shown in Figure 5). The obtained spectral analysis results showed that compared with DAPT alone, the intracellular components changed significantly after combined treatment, indicating that DAPT combined with cisplatin had a synergistic effect on OS cells. Changes in subcellular morphology and biochemical distribution were observed using K-means clustering and univariate imaging. Thus, this study provides a critical understanding of the cellular response of DAPT/cisplatin combination therapy from a biochemical perspective, which provides an experimental basis for exploring therapeutic strategies of other cancer drug combinations in cancer cell systems. D. Gibson et al. [34] reported on the combination of cisplatin and caffeic acid for the treatment of cancer cells. The results show that caffeic acid is a dual-acting drug that can sensitize or habituate cells to cisplatin therapy depending on the time of administration. The co-administration of caffeic acid and cisplatin was found to be effective, which provided a reasonable basis for the preparation of a new platinum caffeic acid compound.

#### 2.1.2. The Second Class of Platinum Drugs

Carboplatin is chemically called cis-1,1-cyclobutane dicarboxylate diammine platinum (II) (shown in Figure 3), and belongs to the second generation of platinum anticancer drugs. The structure is similar to that of cisplatin, where the two NH_3_’s of cisplatin remain unchanged and the two Cl’s are replaced by the chelating coordination of dicarboxylic acid groups. Compared with cisplatin, the clinical therapeutic effect is more significant and the adverse reactions are less severe, and it is soluble in water, so it is very convenient to use [35]. In clinical treatment, the tumors that are treatable with cisplatin can also be treated with carboplatin, but the toxic side effects of carboplatin are small and the symptoms such as nausea and vomiting are mild, so carboplatin has a good application in the clinical treatment of tumors [16]. The anti-tumor mechanism of carboplatin is similar to that of other platinum drugs, in general: after two groups are removed from carboplatin, they are transported to the cell through passive diffusion, attracting water molecules to form positively charged hydrates. By means of electrostatic attraction, the hydrate migrates to the vicinity of DNA, removes two water molecules, and is replaced by the N-7 position on two purine nucleotides to form a platinum-DNA admixture(as shown in Table 1) [36]. Through the combination of these two sites, platinum drugs can form three kinds of cross-linking modes with their DNA, which are intra-DNA cross-linking, inter-chain cross-linking, and DNA–protein cross-linking [37]. The most important cross-linking method is the 1,2-D (GpG) type of in-chain cross-linking, in which the normal expression of DNA is affected, which leads to the apoptosis of tumor cells. In other words, after carboplatin acts on cells, transcription factors that regulate signal channels, such as p38, mitogen-activated protein kinase, extracellular regulatory protein kinase, and stress-activated protein kinase, are activated, resulting in changes in gene expression [38]. Carboplatin forms a six-membered ring with DNA, showing greater stability and water solubility [39]. The metabolism of carboplatin requires two steps (as shown in Figure 6), one of which is the opening of the six-membered ring, a slower but critical reaction process. The second is the dissociation of ligands, which is relatively fast [40]. According to the research results of Yuan Haoyu et al., carboplatin can inhibit p53 or prevent p53 mutation, regulate extracellular protein kinase, reduce its activity, and promote the apoptosis of cancer cells such as cervical cancer [41].

In clinical applications, carboplatin can be used to treat various cancers, such as head and neck cancer, brain cancer, testicular cancer, ovarian cancer, colon cancer, and small cell lung cancer [42]. Carboplatin is a platinum-based derivative of cisplatin with a similar mechanism of action but different structure and toxicity [43,44]. Carboplatin is mainly used in the treatment of advanced ovarian cancer [45]. In addition, its nephrotoxicity, neurotoxicity, and gastrointestinal effects are relatively mild [46], so it can sometimes be used as an effective substitute for cisplatin [47]. Carboplatin is also combined with other drugs to make its clinical effect more effective. When combined with other drugs, it has higher clinical value than simply treating tumors with carboplatin alone, for example, combined paclitaxel treatment of cervical cancer and cervical cancer due to late onset, which is difficult to detect, and often found to be in the late stage of treatment. It has become one of the most common malignant tumors in the female population, having the second most common incidence after breast cancer. Lin Jia et al. [48] have shown that paclitaxel combined with carboplatin chemotherapy and radiotherapy have a significant effect in the treatment of advanced cervical cancer. At present, paclitaxel is the only anticancer drug that can not only block the dissociation of polymerized microtubules, but also promote microtubule merger. When paclitaxel enters tumor cells, its accumulation rate will be accelerated, thus reducing the function of intracellular microtubule tumor cells, blocking the normal division of tumor cells, and ultimately ending the proliferation process of cells during mitosis [49,50]. Carboplatin can also prevent the growth of tumor cells by destroying the structure of DNA molecules. Therefore, the combined use of the two treatment methods has a significant effect on the treatment of cervical cancer in the middle and late stages. In order to improve the efficacy of carboplatin, nanomedical drug delivery has enormous potential. Cytotoxicity, in vitro drug release, and characterization of carboplatin-loaded polybutylcyanoacrylate nanoparticles (PBCA NPs) were investigated by Majid et al. [51]. In this study, PBCA NPs coated with a hydrophilic polymer, polyethylene glycol 3350 (PEG), were prepared by a novel microemulsion polymerization method, which improved the performance of the NPs and the therapeutic effect of carboplatin on ovarian cancer cells. The efficiency of nanomedicine was determined by MTT assay in ovarian cancer cell lines. The zeta potential of NPs is −10.7 mV and the average particle size is 389 nm. The encapsulation rate and drug loading of NPs were 41.43% and 3.59%, respectively. Compared to free carboplatin, NPs have a smaller drug release slope. The results showed that PEGylated NPs had high drug retention ability, the drug release rate was 14% after 38 h, and the drug release rate and the encapsulation rate increased in a time-dependent manner. The results also show that the use of PEG in NP formulations and production processes has a critical impact on NP characteristics, loading rate, and capture efficiency. In addition, the addition of PEG to NP formulations helps improve the performance of NPs and minimizes changes in their properties over time. In PEGylated NPs, the cytotoxicity of carboplatin is associated with a significant increase in drug concentration.

Nedaplatin, also known as cis-glycolate diammine platinum (as shown in Figure 3), is a new second-generation platinum drug developed by Shionogi Pharmaceutical Company in Japan. Nedaplatin has certain curative effect on head and neck tumors, small cell lung cancer, bladder cancer, ovarian cancer, esophageal squamous cell cancer, and cervical cancer, and is widely used in Japan, China, and other countries. After entering the cell, the bond between the alcoholic oxygen on the glycolate ligand and the platinum breaks, and the water binds to the platinum, resulting in the formation of ionic substances (active substances or hydrates). The broken glycolate ligands then become unstable and are released, producing a variety of ionic substances that bind to DNA and inhibit DNA replication, thus generating anti-tumor activity [52]. The dissolution rate of nedaplatin is about 10 times that of cisplatin, and the toxicity is significantly reduced compared with cisplatin, especially with low renal toxicity. Currently, it has been proven that the low renal toxicity is caused by the different distribution of these two drugs in the kidney. When mice are given the same dose of nedaplatin and cisplatin, the cumulative amount of nedaplatin in the kidney is only 40% that of nedaplatin [52,53]. Because there is no need for hydration during use, the application of nedaplatin has been transferred to the outpatient clinic abroad, which greatly assists patients. In past studies, nedaplatin alone or in combination with other drugs has shown good clinical efficacy. In the study of Zhang Pin et al. [54], the total effective rate of nedaplatin in the treatment of various malignant tumors was 12.9%, and there was no complete cross-resistance with cisplatin. The efficacy of nedaplatin as a single drug was similar to that of cisplatin, but the gastrointestinal reaction was less severe than that of cisplatin, and the patients had good tolerance(as shown in Table 1). Chen Jun et al. [55] found that nedaplatin and cisplatin had similar remission rates and total remission rates for primary lesions and cervical lymph node metastases in patients with locally advanced nasopharyngeal carcinoma, with similar efficacy, but that nedaplatin had higher safety, so nedaplatin could be used as an alternative treatment for patients with cisplatin intolerance. Niibe et al. [56] treated locally advanced cervical cancer with radiotherapy and nedaplatin, and the 3-year survival rate of patients reached 73.0% without severe acute or delayed onset of toxic reactions, indicating that nedaplatin was relatively effective and safe. In the study of Lu et al. [57], by comparing nedaplatin and cisplatin in combination with docetaxel for the management of patients with proven advanced SCLC, it was found that the overall treatment response rate and disease control rate of the nedaplatin group were higher than those of the cisplatin group. Although there was no difference in progression-free survival rates between the two groups, the incidence of non-hematological toxic reactions was lower in the nedaplatin group. Therefore, nedaplatin can be used as a new alternative drug for advanced or recurrent squamous cell lung cancer. Also, although the efficacy of nedaplatin is similar to that of cisplatin in monotherapy, the side effects are lower, so nedaplatin can be used as an alternative treatment for patients with cisplatin intolerance. In addition, nedaplatin has a certain synergistic effect when used in combination with other anticancer drugs, which can improve the anticancer effect. Therefore, it is of great significance in clinical application.

#### 2.1.3. The Third Class of Platinum Drugs

Oxaliplatin’s chemical name is (1R,2R)-trans-diamine cyclohexane oxalate platin, also known as oxalate platin, and belongs to the third generation of platinum anticancer drugs. Its structure can be seen as replacing the two amino groups of cisplatin with 1, 2-diaminocyclohexane groups, and the two chlorines with oxalate groups (see Figure 3). Oxaliplatin was originally synthesized by Yidani, developed and developed by Debio Pharm in Switzerland, and first launched in France in October 1996. It was approved by the Food and Drug Administration (FDA) in August 2002 [58]. Our country approved the import of oxaliplatin in 1999, and successfully developed domestic oxaliplatin in 2000 [58]. Although it is a platinum metal drug like cisplatin and carboplatin, the most significant thing is that it does not produce cross-resistance, so the clinical effect is very good when treating carboplatin- and cisplatin-resistant malignant tumors. Moreover, it is also the first platinum metal drug with significant therapeutic effect in the clinical treatment of colon cancer [59]. In addition, it also has a good effect on tumors treated by cisplatin and carboplatin [59]. The oxaliplatin structure contains a 1,2-diaminocyclohexane group, platinum atoms, and a DNA cross-link to produce compounds and prevent DNA repair and replication, resulting in cell apoptosis [60]. Due to the large volume of the compound, the cross-linking between the chains is lower. The high-mobility group has a low affinity for oxaliplatin, and the transcription factor Y-box binding protein can bind to the binding of oxaliplatin to DNA [61]. Under the condition of constant concentration, cisplatin mainly reduces the replication rate and activates the late DNA synthesis (cell cleavage stage), while oxaliplatin activates the early DNA synthesis (replication stage) and blocks the late DNA synthesis (cell cleavage stage) [62]. It has been reported in the literature that among the 117 genes related to oxaliplatin regulation, 79 genes have the same effect as cisplatin, and the other 38 genes are inhibited due to dose dependence [63]. In addition, oxaliplatin can affect the exchange pathway of sodium and calcium plasma, which can lead to neurotoxicity in the body [64]. Oxaliplatin may have a major application advantage in the treatment of metastatic colorectal cancer. It can be used alone or in combination with 5-fluorouracil and leucovorin to achieve a certain effect. In the case that cisplatin and other platinum-based anticancer drugs cannot obtain significant curative effect, oxaliplatin alone has no significant effect in the treatment of colon cancer, but clinical oxaliplatin combined with other drugs can play a certain curative effect. Related research reports pointed out that oxaliplatin and paclitaxel combined use can significantly improve the clinical efficacy in ovarian cancer patients, and that 5-fluorouracil, capecitabine combined treatment of colon cancer, can further improve the clinical effect compared with a single drug. Combined with capecitabine for the treatment of elderly patients with advanced cardiac cancer, the drug can quickly relieve the clinical symptoms of patients, and the adverse reactions are controllable. For relapsed and refractory non-Hodgkin’s lymphoma, dexamethasone, high-dose cytarabine, and oxaliplatin are effective. Cui [65] explores the clinical efficacy of oxaliplatin combined with chemotherapy in the treatment of colon cancer. Oxaliplatin combined with chemotherapy has a significant effect on colon cancer patients. Significantly increasing TNF-αand IL-2 levels can effectively improve clinical symptoms and improve patients’ quality of life. Lan et al. [66] studied the efficacy of oxaliplatin combined with tigio in the treatment of advanced esophageal cancer and its impact on patient survival time. The results showed that oxaliplatin combined with Tigor in the treatment of advanced esophageal cancer patients can improve the level of tumor markers, improve drug safety, and prolong survival time.

The third-generation platinum anticancer drugs also include lobaplatin and heptaplatin (as shown in Figure 3). Among these, lobaplatin, whose chemical name is 1,2-diaminomethylcyclobutane platinum(II) lactic acid (as shown in Figure 3), is a clinically representative platinum complex, successfully marketed in China in 2003. Lobaplatin is characterized by its ability to generate DNA damage through cross-linking of GG and AG chains, thereby forming DNA drug complexes that inhibit tumor activity and further affect the expression of certain specific genes in tumor cells [67]. Studies have shown that lobaplatin has been shown to have good therapeutic effect on some tumors in clinical practice. For example, lobaplatin has been proven to inhibit the proliferation and peritoneal metastasis of colorectal cancer in phase II clinical trials [68], can also inhibit gastric cancer cells by inducing apoptosis [69], can effectively inhibit human non-small cell lung cancer cells in phase S, and triggers apoptosis in human non-small cell lung cancer [67,68,69,70]. In addition, lobaplatin has shown lower drug resistance in many malignancies compared with carboplatin and has lower drug toxicity in patients [71]. Lobaplatin is currently registered in China for the management of metastatic disease in breast cancer and small cell lung cancer SCLC. At the same time, lobaplatin, a chemotherapy drug used in the treatment of chronic myeloid leukemia, has also been widely recognized as effective [72], and some studies have suggested that lobaplatin also has an inhibitory effect on retinoblastoma [73]. The chemical name of heptaplatin is cis-malonic acid [(4R, 5R)-4, 5-bis-(aminomethyl)-2-isopropyl-1, 3-dioxopentyl] platinum (II) (as shown in Figure 5), which is a new generation of platinum-based anticancer drug developed by South Korea’s Sunkyo Pharmaceutical Co., LTD., in 1992. First marketed in Korea in 1999, heptaplatin’s outstanding advantage is that it still has a significant effect on cisplatin-resistant cells or tissues, while its tissue toxicity is lower. Dose-restriction toxicity can be seen in hepatotoxicity, nephrotoxicity, and myelosuppression, and is mainly used to treat advanced inoperable gastric cancer and SCLC [74].

#### 2.1.4. Other Platinum Drugs

In October 2009, Sumitomo Pharmaceutical Co., Ltd. of Japan developed and marketed miplatin, a new fat-soluble platinum complex used primarily to treat liver cancer (see Figure 7). In January 2010, miplatin and its special suspension went on sale simultaneously. This product is an anticancer drug dissolved in special iodized poppy seed oil fatty acid ethyl ester and administered within the hepatic artery. It has a high affinity with the fatty acid ethyl ester of poppy seed oil and remains at the tumor site after intrahepatic arterial administration. The platinum complexes in the suspension can be slowly released into the blood or tissues for a long time and bind to DNA, thus inhibiting the proliferation of cancer cells by preventing DNA synthesis and improving the anticancer effect [75]. Clinical trials have shown that in both patients who have received this treatment for hepatocellular carcinoma for the first time, and some patients who have received other treatments such as liver resection and relapse, this product has shown good anticancer effects, resulting in side effects similar to those of the aforementioned platinum drugs. The side effects of miplatin chemotherapy, which is received in a medical institution that is proficient in this type of therapy, can be controlled within the tolerance range.

Polynuclear platinum is a new type of non-classical platinum anticancer drug; its structure is different from cisplatin and carboplatin, and its research has become a major breakthrough in platinum anticancer drugs. Farrell’s team synthesized a large amount of polynuclear platinum with alkyl diamines as bridging groups. Polynuclear platinum molecules are composed of two or more platinum atoms, which can bind to the DNA of cancer cells through non-covalent effects (electrostatic, hydrogen bonding). Compared with classical platinum complexes, they have a strong aggregation ability, so they will seriously damage the structure and function of DNA, thus making it difficult for tumor cells to self-repair. Therefore, polynuclear platinum complexes have stronger anticancer activity. Clinical studies have shown that BBR-3464 (as shown in Figure 7) has a higher degree of cross-linking than cisplatin in DNA and has strong anticancer activity without cross-resistance [76]. Therefore, it is used to treat some patients with advanced cancer, who often have developed resistance to some typical platinum anticancer drugs.

The Pt(IV) complex is a potential anticancer drug with a higher degree of kinetic inertness compared with the Pt(II) complex. Ji et al. [77] designed a new Pt(IV) complex [Pt(NH_3_)_2_Cl_2_(C_10_H_15_N_2_O_3_S) (C_2_HO_2_Cl_2_)](DPB) (as shown in Figure 7). The effects of the axial ligands on the activity of the complexes and the energy metabolism of tumor cells were also discussed. A number of Pt(IV) complexes are currently under development that are specific to the characteristics of tumor cells, and as Pt(IV) coordination chemistry develops, new complexes will be generated to test whether the molecular targets identified are specific to cancer cells. Although there is great potential for conjugation with axial ligands, the development of synthetic chemistry is able to provide new donors for axial sites to allow for a wider range of reduction potentials and reactivity. Compared with Pt(II) homologues, Pt(IV) precursors have therapeutic advantages. Unlike the square flat Pt(II) complex, the low-spin d6 Pt(IV) centre uses octahedral architecture and is therefore coordinately saturated. Therefore, Pt(IV) precursors are more tolerant to ligand exchange than their Pt(II) counterparts, reducing toxic side effects [14].

Gregory et al. [78] reported new Pt(IV) complexes containing ferrocene (Fc), which differ only in the properties (esters and amides) of the functional groups connected to the Fc subunit. A small structural change (one atom difference) can lead to significant differences in the solubility, stability and anti-proliferative activity of lung cancer cell A549. Huang et al. [79] designed and prepared a novel Pt(IV) complex containing a monoaminophosphate moiety. This complex can not only be used as a bone delivery agent, but also can inhibit the activity of matrix metalloproteinases (MMPs). Compared with cisplatin and oxaliplatin, the compound has stronger antitumor activity while being less toxic to normal human liver cells. In recent years, the use of platinum and other metals to form complex complexes, so as to obtain dual-function or multi-function of new platinum drugs, has become one of the emerging research hotspots. A novel heteronuclear Pt(IV)-Ru(II) anticancer drug was reported. The complex showed good stability in aqueous and PBS buffer solutions. Biological evaluation showed that the bifunctional Pt(IV)-Ru(II) complex has cytotoxic and anti-metastasis properties, taking advantage of the advantages of two metal centers. The tumor-normal cell co-culture system further confirmed its good selectivity for cancer cells [80]. Li et al. [81] prepared dendritic monodisperse Se or platinum coordination dendritic polymers, in which Se is the embedded platinum core. Dendritic macromolecules with coordination dendritic macromolecules show controllable anticancer activity without adding any drugs, and their anticancer activity and low toxicity to normal tissues have been confirmed by in vivo studies. Malgorzata et al. [82] described new Au(III) and Pt(II) complexes containing imine phosphate ligands, and further studies showed that apoptosis is the main mechanism of cell death caused by these compounds. Xiao Hui et al. [83] developed a self-targeting nanoassembly (STNA) using an amphiphilic prodrug of platinum (IV)—lactose for the synergistic and safe radio-chemotherapy of liver cancer. By targeting lactose to liver cancer cells, the platinum STNA can improve tumor aggregation. After receptor-mediated internalization, platinum STNA releases cisplatin (II) in cancer cells to bind to DNA, inducing DNA damage and apoptosis. At the same time, DNA binding also arrests the cell cycle in the radiation-sensitive G2/M arrest phase by upregulating the expression of phosphorylated checkpoint protein 1 (p-Chk1). In addition, platinum STNA, as a radiosensitizing agent, has a strong X-ray attenuation capacity under X-ray irradiation and can store more energy, thereby increasing the level of ROS during the G2/M phase, thereby amplifying the cell-killing effect of radiation therapy in the case of increased DNA damage. The results showed that platinum STNA had no adverse reactions in vivo or in vitro and showed obvious synergistic therapeutic effect in chemoradiotherapy(as shown in Table 1). Taken together, we know that what they propose is a new self-targeting nanoassembly strategy based on widely used platinum drugs that can be used for synergistic chemotherapy and radiosensitized liver cancer therapy.

### 2.2. Ruthenium Anticancer Drugs

In 1980, Clarke reported for the first time the ruthenium metal drugs such as cis-[Ru(NH_3_)Cl_2_] and cis-[Ru(II)Cl_2_(DMSO)_4_], and found that they had very low toxic side effects and strong anticancer activity [84]. Its anti-tumor properties were discovered by Italian chemist Giovanni Mestroni in 1984. Studies have shown that cis-[Ru(NH_3_)Cl_2_] and cis-[Ru(II)Cl_2_(DMSO)_4_] can inhibit the growth of several mouse metastatic tumors [85]. Thus, researchers began to study the anti-tumor properties of ruthenium metal drugs. It has a number of oxidized states, Ru(II), Ru(III), and Ru(IV), which may be regulated by physiological conditions. Ruthenium complexes are suitable alternatives to Pt(II) anticancer agents because they have the same ligand-substitution kinetic profile as Pt(II) complexes in aqueous media. The activity of various ruthenium complexes against malignant tumors has been studied and exploited for several decades. However, the real breakthrough came with the discovery of NAMI-A and KP1019 (Figure 8) [86]. This complex has a significant effect on metastatic malignant tumors and tumors that have become resistant to the classical platinum complex. Although these two drugs are structurally similar, they have different toxicities. NAMI-A has low activity against newly developed tumors but high activity against recurrent tumors [87]. KP1019 has shown strong cytotoxicity to primary tumors, especially in colorectal cancer [88]. It should be noted that both complexes have moderate activity in vitro. However, they are better tolerated in vitro and in clinical practice [89]. Because the results of the phase I trial of NAMI-A showed high cellular toxicity, low efficiency, and a variety of side effects, the drug was not tested further [90]. Due to its low solubility in water, KP1339 is a sodium salt derivative of KP1019 (Figure 8), which is equally effective against colorectal cancer [91]. After extensive research, Sava et al. found that NAMI-A can selectively de-attack potentially metastatic cancer cells [92]. NAMI-A binds to collagen in the extracellular matrix and to integrins on the cell surface, increasing the adhesion and decreasing the invasiveness of cancer cells. In contrast, the mode of action of KP1019 and KP1339 is completely different from that of NAMI-A. KP1019 and KP1339 are prototypical cytotoxic drugs that follow the mitochondrial apoptotic pathway, interacting strongly with mitochondria and DNA, leading to ROS, oxidative stress, and endoplasmic reticulum damage, which in turn induces the process of apoptosis [93]. There is no exact conclusion on the mechanism of action of ruthenium anticancer drugs, because ruthenium metal drugs have various structures, so drugs with different structures have different reaction mechanisms. Studies in vivo and in vitro have shown that NAMI-A and many ruthenium metal drugs have almost no toxicity, and the ligands of ruthenium metal drugs can be hydrolyzed and dissociated under certain conditions, and the hydrolyzed products have stronger nucleophile ability and are easier to bind to the target molecules in vivo. Therefore, the rate of hydrolysis will directly affect the anticancer activity of such metal drugs. The target molecule of ruthenium anticancer metal drugs in vivo is usually DNA, but there are three ways of binding anticancer drugs to DNA, namely electrostatic binding, insertional binding, and furrow binding. Many ruthenium drugs have a positive charge, so they can be electrostatically bound to DNA. Ruthenium drugs can also be combined with DNA insertions, where they act as inserters. It can also cross some base pairs to bind to small and large furrows in the DNA double helix structure [94].

#### 2.2.1. Ruthenium(II) Arene Complexes

In 1992, Tocher first reported the antitumor activity of arylruthenium (II) complexes [95]. Sadler and Dyson were pioneers in the study of arylruthenium anticancer complexes [96,97]. The arylruthenium (II) complex is an organometallic compound with a “piano stool” structure, [(η^6^-arene)Ru(X)(Y)(Z)] (Figure 9, **1**), with a unique octahedral geometry with low d^6^ spin values, with Ru as the central ion and aromatic “cap” of the aromatic ligand as the “stool surface” occupying three coordination sites. Common aromatic rings include benzene (ben), methylisopropylbenzene (cym), biphenyls (bip), and dihydrophenanthrene (dha). The remaining three ligands are occupied by the other ligands X, Y, and Z (or “stool legs”) [95,98]. Ligands X and Y can be either monodentate or bidentate ligands (N^N, N^O, O^O, S^O, etc.), and Z is usually the leaving group, such as the halogens F, Cl, Br, etc. [95,99]. The aromatic ring is considered to be the core component of the arylruthenium Ru(II) complex, which determines the distribution of the electrons in the Ru(II) complex and thus the stability of the Ru(II) complex. The aromatic ring is lipophilic, which is conducive to the entry of Ru(II) complex into cells, and the greater the hydrophobicity, the greater the cytotoxicity. According to literature reports, the larger the size of the aryl group, the stronger the interaction between the complex and the base, and the stronger the anticancer activity [100]. In addition, the hydrolysis rate of Ru-Z bond is also affected by the pH and Z ion concentration in the environment [101]. And the change of ligand can affect the cytotoxicity and stability of the complex. After hydrolysis of the leaving group, the Ru(II) complex produces an empty coordination site, which facilitates the interaction of metallic Ru(II) with biomolecules. The hydrolysis of the leaving group contributes to the cytotoxicity of metal antitumor drugs, but the rate of hydrolysis of the leaving group is not linearly positive with the antitumor activity of metal. The ideal state is that when the leaving group is hydrolyzed and the complex just reaches the target, which can avoid the toxic side effects caused by the combination of the complex with other biomolecules. Therefore, it is important to introduce appropriate groups to control the rate of hydrolysis of the complex [102]. Studies have shown that with the arylruthenium complex [(η^6^-arene)Ru(XY)Z]^+^, the more stable Z is, the slower the hydrolysis rate of the complex, and vice versa. When Z is a halogen atom (Br or Cl), the hydrolysis rate is faster. When Z is imidazole or pyridine, the hydrolysis rate is slower. The activity of the complex also has an important relationship with the ligand, so the antitumor activity of the complex can be further regulated by introducing a different ligand [98]. Phosphorus, nitrogen, and sulfur are common monodentate ligands; N^N, O^O, and N^O are common bidentate ligands. Dyson et al. [103] reported Rapta-type compounds (Figure 9, **2**–**12**), on the aryl Ru(II) complex of 1,3, 5-triaza-7-phosphoheteradamantane (PTA) ligand; the general structure formula is [(η^6^-arene)Ru(X)(Y)(PTA)]. Arene is methyl isopropylbenzene (p-cymene), toluene, benzene, hexamethylbenzene, etc., of which X and Y are most commonly chlorine [104]. The hydrophilic PTA ligands have good water solubility and preferentially protonate in low-pH environments. Chlorinated RAPTA derivatives are readily hydrolyzed in low-chloride-ion environments [105]. Keppler et al. [106] designed RAPTA-C analogized PTA with glucofuranoside ligands by adjusting the lipophil properties of the complexes, and obtained six ruthenium complexes containing glucose molecules with high water solubility suitable for intravenous administration (Figure 9, **13**–**18**). Six complexes were tested for antitumor activity in vitro, and the results showed that compound 18 had good lipophilic properties, which were conducive to cell absorption, and the strongest anti-tumor activity (human chordoma CH1, IC50 = 29 µmol/L). Fabio’s research group [107] prepared and characterized a series of arylruthenium (II) complexes (Figure 10, **19** and **20**), which were connected to the metal center by biotin-functionalized triphenylphosphine ligands to synthesize a series of arylruthenium (II) complexes containing biotin fragments. It shows excellent stability in a dimethyl sulfoxide/water mixture. The synthesized biotin conjugated arylruthenium Ru(II) complex can be efficiently taken up by tumor cells with enhanced cellular uptake and selectivity. In addition to single-tooth P-ligands, imidazole, pyridine, triazole, and quinoline derivatives are commonly used in the synthesis of aryl ruthenium complexes as common single-tooth nitrogen ligands. In order to overcome the shortcomings of multi-drug resistance in tumors, Dyson and Vock et al. [108] synthesized phenoxazine and anthracyl complexes with imidazole as ligands by introducing anti-drug resistant molecules that inhibit P-glycoprotein (P-gp) and imidazole as bridging ligands. It has been reported that coumarins also have good anti-tumor activity [109]. Zhao et al. [110] designed and synthesized three arylruthenium (II) complexes containing 7-hydroxycoumarins by pyridine bridging. The MTT assay demonstrated that the synthetic arylruthenium (II) complexes exhibited higher cytotoxicity to cancer cell lines than ligands. In coordination chemistry, acylthiourea compounds have a variety of coordination modes due to the presence of S, N, and O in the structure as electron-donating groups, one of which is the coordination of sulfur ligands with metals. Cunha et al. [111] synthesized a series of arylruthenium (II) complexes containing acylthiosemide ligands with the structural formula [Ru(η^6^-p-cymene)(PPh_3_)-(S)Cl]PF_6_ (Figure 10, **21**–**26**). Five cell lines were selected, including A549, MDA-MB-231, MRC-5, MCF-10A, and DU145, and the activity of these compounds was evaluated by MTT assay. Compared with cisplatin, these complexes showed higher cytotoxicity to MDA-MB-231 (IC50 = 0.28~0.74 μmol/L) and A549 (IC50 = 0.51~1.83 μmol/L), and their activity was significantly higher than that of cDDP. A significant SI (4.66–19.34) was found for breast cancer cells.

The common N-immediately N-binodentate ligands include aliphatic diamines, aromatic diamines, and pyridine derivatives. The cytotoxic mechanism of this arylruthenium (II) involves the hydrolysis of Ru-X bonds to produce active Ru-OH_2_ [113]. The complex will be present in a certain pH range, but above the pH = pKa value (50% in the form of Ru-OH_2_ and Ru-OH via deprotonation of the H_2_O ligand), the hydroxyl Ru-OH will dominate, and the complex is generally considered to be less active. Hydroxide is not as easily exchanged by biomolecular targets because it is a more stable ligand than water [113]. Ideally, therefore, a pKa of around pH 7 for aqueous adducts should ensure that the active ingredient is dominant at physiological pH (7.2–7.4). Sadler’s group [113] studied a series of ethylenediamine arylruthenium (II) complexes such as [(η^6^-p-cymene)Ru(en)Cl]^+^ (Figure 10, **30**), [η^6^-biphenyl)Ru(en)Cl]^+^ (Figure 10, **31**), [(η^6^-biphenyl)Ru(en-et)Cl]^+^(Figure 10, **32** and **33**),[(η^6^-biphenyl)Ru(en-et)Cl]^+^ (Figure 10, **34**); the general formula is [(arene)Ru(en)Z]^+^. All of these agents inhibit the growth of A2780 cancer cells with IC50 values between 6 and 9 µmol/L, comparable to the commonly used anticancer drug carboplatin (IC50 = 6 µmol/L). In addition, complexes involving more hydrophobic moieties such as [η6-tetrahydronaphthalene)Ru(en)Cl]+ have higher activity, with IC50 values comparable to cisplatin (0.6 µmol/L) [114]. The structure–activity relationship (SAR) showed that the introduction of polar substituents to the coordination benzene ring of the [(η^6^-arene)Ru(en)Cl]PF_6_ complex reduced their cytotoxicity to A2780 human ovarian cancer cells [114]. RM175, RAED is a typical ethylenediamine arylruthenium (II) complex [115]. RM175 has high cytotoxicity both in vivo and in vitro. The anticancer activity of RM175 has similar activity to carboplatin in the human ovarian cancer cell line A2780. It can also effectively overcome cisplatin resistance. Aromatic substituents help the complex enter the cell by increasing its lipophilicity, and when the complex is hydrated, it covalently binds to the N7 guanine of DNA [116]. The hydrophobic interaction of RM175 (Figure 10, **32**) with DNA was achieved by the extension of the aromatic ring into the base pair. Experiments have shown that RM175 [98] can inhibit the growth of primary MCa breast cancer and lung metastases. Moreover, increasing the lipophilicity of the aryl group can improve the cytotoxicity of the complex. In addition to binding to DNA, RM175 also inhibits matrix metalloproteinase-2 (MMP-2). MMP-2 can inhibit the adaptive immune system and play a key role in tumor penetration and spread by removing the extracellular matrix (ECM) and destroying the histological barrier of cancer cell invasion. It controls the tumor microenvironment and regulates cell growth and angiogenesis through different signaling pathways. Chloride ions replaced the hexafluorophosphate (PF^6−^) anion of RM175 to form ONCO4417 (Figure 10, **33**) [117], showing good antitumor activity and G2/M phase cell cycle arrest. The complex has been shown to have similar potency to cisplatin in a variety of cell lines (ovarian, lung, oesophageal, pancreatic, melanoma, colorectal, etc.). RAED can bind to DNA and form an adduct with guanine [118]. Changes in the aromatic ring affect the biological activity of the complex. For example, the more hydrophobic complex of the aromatic ring ligand (tetrahydroanthracene) (Figure 10, **31**) is 10 times more cytotoxic than that of the aromatic ring ligand (methyl isopropyl benzene) (Figure 10, **30**). According to literature reports, the arylruthenium (II) complex with two chlorines as ligands is easily hydrolyzed, while the O^O bidentate ligand is relatively stable, which inhibits the hydrolysis of the complex to a certain extent, because it can form a stable six-membered chelate structure after coordination with metallic ruthenium, thus improving the therapeutic effect of drugs [119]. Cyclooxygenase (COX, specifically COX-2) is commonly up-regulated in malignant tumors, and COX-2 is a key link in triggering subsequent inflammatory responses. NSAID (non-steroidal anti-inflammatory drug) is a kind of drug with anti-inflammatory, anti-rheumatic, analgesic, antipyretic and anticoagulant effects; NSAID mainly targets the cyclooxygenase (COX-1 and COX-2), and inhibits the production of prostaglandins. These NSAIDs have the effect of inhibiting epidermal growth factor (EGF) function and overexpression of tumor suppressor gene activity. It has been reported that NSAIDs have a synergistic effect with anti-tumor drugs [120] and can cause apoptosis. Chanchal et al. [121] selected three different NSAIDs as ligands, reported three areneRu(II)-NSAID complexes (Figure 10, **27**–**29**), and systematically studied their anti-proliferation and anti-metastasis activities. The results showed that the areneRu(II)-NSAID complex had a significant inhibitory effect on tumor metastasis, and could induce cell apoptosis and block the cell cycle in the G2/M phase. It has been shown that the embedded binding mode of the aromatic moiety is very helpful in unwinding double-stranded DNA and allows the drug to work in combination with DNA. Some arylruthenium complexes act through S-phase cell cycle arrest by DNA damage-mediated phosphorylation of p53. p53 is a transcription factor that triggers a variety of stress signals such as DNA damage, oncogene activation, and so on. Cell cycle arrest and cell death are among the major results of p53 activation. p53 is activated and the cell cycle is first arrested at the G1 checkpoint phase by activation of CDK inhibitor (CKI) proteins. The CDK inhibitor then binds to the CDK-cell cycle protein A complex, blocking S phase cell cycle activity and initiating the process of apoptosis [122].

#### 2.2.2. Ruthenium(II) Polypyridine Complexes

Photodynamic therapy (PDT) is a non-invasive technique used clinically to treat a wide range of cancers and bacterial, fungal, or viral diseases [123,124]. Among the novel PSs studied, Ru (II) polypyridinyl complexes have excellent excited state properties, high water solubility, high chemical stability, and light stability, strong luminescence, large Stokes shift, and biocompatibility, so they are widely used in PDT [125,126,127]. Chao et al. proposed a series of novel asymmetrically substituted 1,10-phenoline and 4,7-diphenyl-1,10-phenoline Ru-II polypyridine complexes **35**~**38** (shown in Figure 11) as effective photosensitizers for single-light and two-photon PDT. The complex shows a redshifted single-photon absorption in the biospectral window and has a very strong two-photon absorption. They enter cells through an energy-dependent endocytosis mechanism, and after irradiation in various 2D monolayer cancer cell lines, the complex apoptoses through the caspase 3/7 pathway, resulting in cell death in the high nanomolar/low micromolar range [128]. Liu et al. in 2010 prepared two Ru compounds of the type [Ru(dmb)2(DBHIP)](ClO4)2 and [Ru(dmp)_2_(DBHIP)](ClO_4_)_2_ (Figure 11, **39** and **40**) with a new ligand 2-(3,4),5-dibromo-4-hydroxyphenyl)imidazole and [4,5-F][1,10] phenanthroline (DBHIP). These complexes have excellent optical properties. They facilitate the study of the binding of DNA and ethidium bromide by emission spectroscopy. Insertion binding to DNA has also been reported. The mechanism of action involves the formation of singlet oxygen (^1^O_2_) and superoxide radical anion (O^2−^). These radicals are involved in DNA cleavage. These compounds have a good antioxidant capacity, are able to lyse plasmid DNA, and have a high level of cytotoxicity to tumor cell lines [129].

Polypyridine-ruthenium(II) complexes have become a new type of photosensitizer because of their highly selective action on tumor cells and their toxicity to any healthy cells. Chen et al. [131] designed a series of Ru polypyridine complexes (RUpops) in 2022 to systematically explore their potential in promoting NK cell therapy (anticancer mechanism shown in Figure 12). Interestingly, its structure can determine the anticancer activity of Ru complexes, whereas only polypyridine ruthenium can effectively modulate immunosuppressive factors and target proteins in tumor cells. This unique property contributes to the patient’s MDA-MB-231 cell sensitivity to NK cells. In addition, RuPOP co-acts with NK cells; in addition to directly damaging tumor cells, it can also trigger CaspasE3-dependent apoptosis by upregulating NKG2D and its multiple ligands, induce ROS production, and activate a variety of apoptosis-related receptors, such as tumor cells and tumor cells, so as to maximize the interaction between NK cells and tumor cells(as shown in Table 1). Wang Yan’s research group [132] synthesized a lysosomal targeted polypyridyl ruthenium (II) and synthesized a lysosomal targeted polypyridyl Ru (bpy)_3_(bpy=2,2′-bipyridine, rhein=4,5-dihydroxy-9,10-dioxoanthracene-2-carboxylic acid), and connected with Chinese herbal rhein. The complex showed strong short-term phototoxicity against cancer cells relative to cisplatin, with IC50 values in the range of 2.4–8.7 μM. In addition, it has low toxicity to cells in dark environments, which helps to reduce its toxic side effects on normal cells. The experimental results indicate that this complex can induce cancer cell death through the autophagy pathway. Therefore, lysosome-targeted photosensitizers based on Ru compounds have great potential for the application of PDT in cancer treatment. The Dipankar Nandi research group [133] prepared two ruthenium (II) complexes [Ru (tpy BODIPY)_2_] Cl_2_ (tpy=4-phenyl-2,2: 6,2-tripyridine, BODIPY=boron dipyrrolidene), and studied their phototherapeutic activity and biological imaging performance. Complexes with structural similarity only affect their physical and chemical properties due to differences in phenylethyl ligands. These two compounds show high absorption around 500 nm in DMSO (ε: ~1.5 × 10^5^ M^−1^ cm ^−1^), high quantum yield of singlet oxygen and low bleachability, making them useful for PDT applications. Both complexes have high DNA binding efficiencies and can generate various types of ROS and induce photoactivated DNA damage.

### 2.3. Iridium Anticancer Drugs

Platinum-based anticancer drugs, although successfully used in clinical treatment, have strong toxic side effects, so researchers have begun to explore other metallic anticancer drugs [134]. In the last decade, the interaction of iridium(III) complexes with biomolecules has been studied. Binding to oligonucleotides, amino acids, peptides, and proteins was mainly studied [135,136,137,138,139]. In recent years, the activity of iridium(III) compounds against cancer cells has sparked interest, which has led to the design of iridium-based anticancer drugs. Ir(I) compounds such as the mononuclear [Ir(acac)(cod)] [137] and binuclear [IrCl(cod)]_2_ (shown in Figure 13) [140,141] have planar tetragonal space structures similar to cisplatin. They have been shown to be remarkably effective in suppressing tumors and metastasis in vitro and in vivo. Ir(III) reagents have been used in anti-proliferative assays to study cancer cells and have demonstrated a strong potential for application. In addition, Ir(III) reagents can be used as imaging probes for the early detection and diagnosis of cancer. Ir(III) is a metal complex with important applications, which has a more stable oxidation state and higher coordination number compared with Ir(I), which gives it superior properties in some specific chemical reactions. At the same time, Ir(III) complexes can also provide a wide diversity of different ligands, which offers more possibilities for chemical synthesis. In the biomedical field, Ir(III) complexes also show many advantages. Firstly, its colour can be changed by simple adjustments, which makes it easier to observe and track its behaviour in experiments. Secondly, energy level control is also an important property of Ir(III) complexes, which allows us to perform chemical reactions or biological experiments at specific energy levels. In addition, the lifetime of Ir(III) complexes is also very long, and can reach the microsecond level, which allows us to observe biological processes more accurately in our experiments. In addition to this, Ir(III) complexes can allow signal recognition of protein autofluorescence. This is because Ir(III) ions can bind to protein molecules, thus allowing them to emit fluorescent signals at specific wavelengths. This method of signal recognition is important for detection and observation in biological experiments. Finally, under low oxygen conditions, Ir(III) complexes can generate ROS by electron transfer (type I) or energy transfer (type II). This property allows Ir(III) complexes to play an important role in specific biological processes, such as in the treatment of some diseases. The unique chemical properties of iridium metal provide a basis for the study of novel diagnostic agents and drugs with new mechanisms of action. Most of the current reports on iridium(III) complexes are of aryl and cyclic metal structures. Most of these studies have focused on the synthesis and application of neutral and cationic complexes [142,143,144,145,146,147,148]. For example, Sadler’s group has prepared neutral and cationic half-sandwich iridium(III) complexes containing C^N bidentate ligands and N^N bidentate ligands, respectively. It was found that the replacement of the neutral N^N bidentate ligand bipyridine (bpy) with the negatively charged anionic C^N bidentate ligand 2-phenylpyridine (phpy) in half-sandwich iridium(III) complexes improves biological activity [143,149,150,151,152,153]. Because iridium-based anticancer drugs are iridium-containing metal complexes, they can be divided into the following two main categories due to the difference in the groups connected to iridium: aryl complexes and cyclic metal complexes. Therefore, they are divided into aryl iridium anticancer drugs and cyclometalated iridium(III) anticancer drugs for presentation in the following.

#### 2.3.1. Aryl Liridium Anticancer Drugs

Iridium is an extremely scarce noble metal with a 5d^6^ outer electronic structure in the +3 valence state. Although iridium(I) complexes have emerged as potential catalysts in a number of hydrogenation reactions, the potential of iridium(III) complexes has not been fully explored and exploited in the pharmaceutical industry [143]. Iridium complexes are primarily considered inert and have a very low ligand exchange rate, but experimental results show that this rate is increased many times with the introduction of cyclopentadienyl ligands (often referred to as Cp*) [152,154]. In 2014, Liu’s team engineered a class of half-sandwich cyclopentadienyl iridium(III) compounds with the general formula [(η5-Cp*)Ir(L∧L’)Z]0/n+ (shown in Figure 13). The Cp* motif allows the coupler to undergo an extension, which helps the compounds insert more easily between DNA strands and interact with the bases. And excellent cell release is provided by the hydrophobicity of the Cp* moiety. These drugs have been reported to have higher efficacy than the clinical drugs cisplatin and oxaliplatin. Depending on the substituent, these complexes exhibit variable ranges of cytotoxic activity against a variety of cancer cells [143]. In 2015, Millett et al. reported 15 semi-filled iridium complexes [(η^5^-Cp*)Ir(2-R’-phenyl)-R pyridine)Cl] (as shown in Figure 13). It has been reported that these complexes bind to guanine better than adenine. This complex showed significant cytotoxicity against human colon, lung, ovarian, and breast cancer cells, with the electron-donating methylphenyl being the most toxic. It is reported that there is a sharp contrast between these structurally similar isomers. Because of their enhanced hydrophobicity, these compounds accumulate in a variety of cells, growing their potency [155].

#### 2.3.2. Cyclometalated Iridium(III) Anticancer Drugs

Iridium compounds have very potent anticancer effects against cancer cells. These complexes are very effective in sensitizing basal oxygen (^3^O_2_) to monoclinic oxygen (^1^O_2_), which is extremely cytotoxic, making these complexes great photosensitizers. Cell division in cancer cells begins with the generation of ROS or singlet oxygen [156]. Cyclometalated iridium(III) complexes also exhibit outstanding biological photophysical properties, such as large ligand division energy, long luminescent lifetime, large displacement, good cell penetration, and cell stabilization, making them great candidates for biological image processing [157]. They also have good biosensing properties, high quantity yield and high photobleaching resistance. They also inhibit protein levels by targeting specific organelles [158]. All these characteristics together make the cyclometalated iridium(III) complex, which is an effective anticancer drug. Mukhopadhyay et al. [159] studied the effects of different substitutions in cyclometalated iridium (III) complexes on DNA and protein targeting and variations in anticancer potential. The team synthesized p-pyridine-based quinolone ligands and attached iridium complexes with of the formula [Ir(ppy)_2_(L)]^+^ PF^6−^. The absorption spectra of these compounds shows strong π-π* transitions at 300 nm and weaker MLCT transitions at 380 to 390 nm. These compounds emit light at wavelengths ranging from 430 to 571 nm when excited. Various experimental studies have shown that these complexes efficiently bind to calf thymus DNA by groove and electrostatic binding to the hydrophobic part of human serum albumin. These complexes also showed considerable anti-proliferative activity against MDA-MB-231 breast and cervical cancer cell lines(as shown in Table 1).

### 2.4. Gold Anticancer Drugs

The absorption spectra of Au^3+^ compounds shows the same strong π-π* transitions at 300 nm and weaker MLCT transitions at 380 to 390 nm^8^ closed shell electronic structure as Pt^2+^, and can form a square-planar structure. These same characteristics make gold a promising anticancer drug to be developed. The early use of the antiarthritic drug kinloprofen has been re-studied and found to inhibit the activity of tumor cells in vitro [160] and in vivo on the mouse P388 leukemia cell line [161]. Therefore, in recent years, scientists have focused their interest on the cytotoxicity and anticancer effects of gold (III) complexes. Of course, Au^3+^ also has its own shortcomings. The most influential point is that Au^3+^ is prone to instability in the physiological environment and can easily be reduced to Au+ or Au(0), and the ligand will also leave, which leads to the loss of efficacy of Au^3+^ before it reaches the target [162]. In order to solve this fatal shortcoming, scientists have made a large number of studies, and it has been shown that simple N-containing binodentate or multi-dentate ligands [163] or other cyclic aromatic ligands can provide enough stability to electrophilic Au^3+^ centers to be stable in physiological environments full of reducing substances for more than 24 h. According to numerous experimental phenomena, gold (III) complexes can effectively bind cisplatin-resistant cancer cells [164], which indicates that the mechanism of Au^3+^ complexes against malignant cell proliferation may be different from that of cisplatin. Moreover, its target is not single; it can be divided into Se or thiolated proteins (cysteine, thioredoxin reductase TrxR, glutathione reductase GR), tyrosine, histidine, DNA, and mitochondria [165,166]. Spectroscopic instruments have detected the interaction of gold (I) complexes with proteins to form SCys-Au single bonds or ScYS-AU-Scys disulfide bonds, which further indicates that Au-S has a strong binding effect. Saggioro’s group studied the cytotoxicity and effects of [Au(DMDT)X_2_][X=Cl, Br] and [Au(ESDT)X_2_][X=Cl, Br] (as shown in Figure 13) on mitochondrial function at cellular and subcellular levels, and found that these two substances can change the internal and external potential of the mitochondrial membrane and thus alter osmotic pressure; stimulate the production of reactive oxygen species; and especially, inhibit the activity of TrxR containing selenium. It can also increase the phosphorylation of ERK1/2. TrxR is the target of anticancer agents, but glutathione reductase, which belongs to the pyridinucleotide REDOX reductase family with TrxR, is not affected, possibly because TrxR contains Se to catalyze the reaction(as shown in Table 1) [167]. Liao et al. [168] have conducted theoretical studies on the reactivity of these two gold (III) complexes containing DMDT and ESDT ligands with purine bases or sulfur-containing proteins, and the results obtained are the same as the actual ones, showing that the S site is a target of Au^3+^. Many reports have reported that organometallic complexes are easy to bind to DNA. These metal complexes have some common pointsand are easy to prepare; a wide variety of complexes can be designed with different ligands; and the binding process is easy to explore with instruments. Navarro et al. [169] synthesized [Au(dppz)_2_]^3+^ (as shown in Figure 13) and experimentally proved that it has a strong toxic effect on Leishmaniasis eggs, and spectrotitration found that it can change the viscosity of DNA, so it is speculated that the cytotoxic effect of this type of gold (III) complex may be related to DNA. Gabbiani et al. designed a series of different OXo-bibritine (III) complexes and studied their inhibitory effects on 36 human cancer cell lines. Spectroscopic and ESI MS data show that different Auoxo complexes have different reaction mechanisms, and some of the biased targets are active protein kinase C in histone deacetylase. Others bind calf thymus DNA significantly, producing irreversible Auoxo/DNA adducts. Kaps et al. carried out a detailed physical property test and cellular toxicity experiment on Au (I)—NHC (i.e., N heterocyclic carving) complexes [170]. The Rana group [171] designed and synthesized a +3 valence Au(III)-NHC complex containing two Cl^−^ and two NHC ligands which is cytotoxic to HCT 116, HepG2, A549, and MCF7 cell lines, which may be due to the inhibition of TrxR by Au(III)-NHC, which disrupts important antioxidant protection systems in cells. As a result, a large amount of ROS was not consumed in time; at the same time, mitochondrial membrane potential decreased and osmotic pressure changed. In general, gold (III) complexes have anticancer potential, but due to insufficient research models, low detection technology, and the influence of the diversity of ligands, the reaction mechanism between them and the target molecules is not clear.

**Figure 13 pharmaceutics-15-02750-f013:**
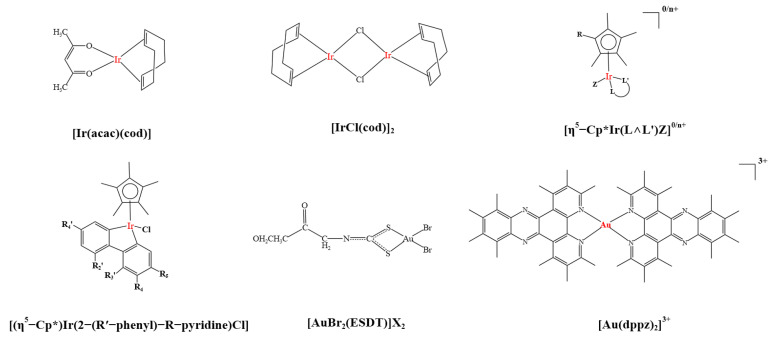
Structural schematic diagram of iridium and gold complexes.

### 2.5. Other Anticancer Drugs

#### 2.5.1. Other Metal Anticancer Drugs

Osmium complexes have rarely been developed for their therapeutic potential due to their toxicity (OsO_4_) and replacement inertia (Os(II) and Os(III) complexes). Sadler et al. fine-tuned the kinetics and thermodynamics of the reaction of osmium complexes in aqueous solution by systematically changing the chelating ligands, so that the aryl osmium (II) complexes showed excellent cytotoxicity to cancer cells compared with the clinical drugs carboplatin and cisplatin [172]. Jana Hildebrandt et al. [173] reported on various ruthenium (II) complexes and promising osmium (II) analogues, all of which use cinnamic acid as an O, S bidentate ligand. The anticancer activity and ability to evade platinum resistance mechanisms of these compounds were studied. The results showed that Ru (II) and Os (II) compounds (as shown in Figure 14) containing O and S bidentate ligands have high cytotoxicity, but their effects on DNA damage and the cell cycle are not strong. This may be the basis for bypassing drug resistance mechanisms and the high specificity of cancer cells [174]. Due to the similarity of the extranuclear electron configuration and chemical properties of isometal elements, in addition to the transition metals platinum, ruthenium, iridium, osmium, and other complexes described above, rhodium complexes in the same group VIII and in the same column with iridium have gradually entered researchers’ field of view. Some studies have found that some Rh (III) complexes can bind specifically to the biological macromolecules DNA and RNA [175]. Metal Rh (III) complexes can recognize the mismatching of guanine (G) and uracil (U) bases on RNA and the action site of a single guanine, and can be cut off under light-induced conditions for the specific recognition of mismatching bases on RNA, opening up a new road for the development of anticancer chemotherapy drugs [176].

**Table 1 pharmaceutics-15-02750-t001:** Types of anticancer drugs, reaction principles and outcomes.

Type of Drugs	Reaction Principle	Application Areas and Results	References
splatin	By the electrostatic attraction of DNA, it is directed to migrate rapidly towards the cell nucleus and reach the target, which alters the function of the normal DNA replication template and affects the division of cancer cells.	Mainly effective in lung cancer, oesophageal cancer, breast cancer, stomach cancer, malignant lymphoma, ovarian cancer, head and neck tumours, etc.	[15,16]
carboplatin	After carboplatin acts on cells, transcription factors that regulate signal channels, such as p38, mitogen activated protein kinase, extracellular regulated protein kinase, and responsive protein kinase, are activated, leading to changes in gene expression.	Used to treat various cancers, such as head and neck cancer, brain cancer, testicular cancer, ovarian cancer, colon cancer, and small cell lung cancer.	[32,36]
oxaliplatin	The structure of oxaliplatin contains a 1,2-diaminocyclohexane group, and the platinum atom crosslinks with DNA to produce a compound that prevents DNA repair and replication, leading to cell apoptosis.	It also has good therapeutic effects on tumors that can be treated with cisplatin and carboplatin	[53,54]
(STNA)	After receptor-mediated endocytosis, platinum STNA releases cisplatin (II) in cancer cells, forming DNA binding and inducing DNA damage and cell apoptosis.	Used for synergistic and safe radiotherapy and chemotherapy of liver cancer. Due to the targeting effect of lactose on liver cancer cells, platinum STNA can improve tumor aggregation.	[83]
RuPOP	RuPOP interacts with NK cells, not only directly damaging tumor cells, but also triggering caspase-3 dependent cell apoptosis by upregulating NKG2D and its multiple ligands, inducing ROS production, activating various apoptosis-related receptors, such as tumor cells and tumor cells, thereby maximizing the interaction between NK cells and tumor cells.	RuPOP effectively regulates immunosuppressive factors and target proteins within tumor cells. This unique property contributes to its good ability to enhance the sensitivity of MDA-MB-231 cells to NK cells from cancer patients.	[131]
Cyclometalated iridium(III) complexes	These complexes bind efficiently to calf thymus DNA by groove binding and electrostatic binding to the hydrophobic portion of human serum albumin.	These complexes also showed significant antiproliferative activity against MDA-MB-231 breast cancer cell line and cervical cancer cell line	[159]
[Au(ESDT)X_2_]	The ability to change the electrical potential inside and outside the mitochondrial membrane thereby altering the osmotic pressure and stimulating the production of reactive oxygen species.	Particularly inhibits selenium-containing TrxR activity; also increases phosphorylation of ERK1/2.	[167]

Rhenium complexes have long been considered potential anticancer agents, and it is only recently that many studies have emerged demonstrating their effective anticancer activity. They are easy to synthesize, can produce a large number of various compounds, and can adjust the performance of and optimize the biological activity. The rich spectral properties of rhenium complexes can be used for fluorescence and vibration microscopy imaging in vitro. The stable Re (I) tricarbonyl nuclear structure is most commonly applied to biological systems [177]. In addition to tricarbonyl Re(I) complexes (as shown in Figure 14), the anticancer properties of rhenium complexes with larger oxidation states have been investigated, focusing mainly on metal ligands and metal–metal multi-bonded rhenium complexes [178]. The study of rhenium in cancer therapy is in its infancy and the mechanism of action of these complexes is still widely unknown and remains to be explored.

Tin complexes (IV) attracted the attention of scientists after their antitumor activity was discovered in 1980, and a large number of tin anticancer drugs were developed [179]. A large number of studies have shown that some tin complexes are of great significance in the treatment of lymphatic leukemia, and their therapeutic effect is obviously better than that of platinum anticancer drugs. However, disadvantages such as a narrow anticancer spectrum and significant toxic side effects seriously hinder their anti-tumor ability. In the 1990s, dibutyltin rutinate compounds were synthesized, with good anticancer activity against MCF and other cancer cells. Since it was found in the 1980s that palladium metal complexes with similar structure to platinum had good anticancer activity against S-180 sarcoma cells, a variety of planar palladium metal complexes have appeared successively, most of which are soft alkali chelated ligands, and the coordination atoms are often N, P, and S [180]. The mechanism of interaction between metal palladium and DNA was studied by the fluorescence method. Gel electrophoresis and transmission electron microscopy confirmed that the metal palladium complex can induce apoptosis and has good anti-tumor activity. It is well known that silver metal has an antibacterial effect, and the antibacterial ability of silver nanoparticles is even stronger. Its good chemical properties have aroused the interest of researchers, and it is found that silver metal complexes have good anti-tumor ability in the process of continuous exploration [181]. Silver-NHC complexes (as shown in Figure 14) have been found to be able to cause the death of breast cancer cells and have good anticancer activity. In addition, the anticancer activity of silver-NHC complexes on cervical cancer cells and lung cancer cells is stronger than that of cisplatin, but it can promote apoptosis of cells without causing significant side effects on normal liver cells. Such metal anticancer complexes have broad prospects in the field of anticancer research. Hessam Hosseinkazemi et al. [182] reported on the potential use of iron oxide nanoparticles as therapeutic drugs and acknowledged their superiority over traditional therapeutic drugs. Although only used for the treatment of iron-deficiency anemia and cancer, multiple experiments are currently being conducted to promote its clinical application. As an essential trace metal element in human growth and development, copper plays an important role as a catalytic cofactor in the physiological processes of cells and participates in the REDOX process of mitochondria [183]. Some copper (II) tetrahedral coordination complexes have anti-tumor activity, and researchers have found that the mechanism of apoptosis caused by copper complexes with different ligands is binding of front-end DNA, resulting in cell growth cycle obstruction. In addition, copper complexes can also cause ROS to accumulate in large quantities in cells, leading to apoptosis [184]. As the first non-platinum anticancer metal drug to enter clinical research, researchers have great expectations for titanium complexes [185]. Titanocene dichloride, developed in the 1970s, has anticancer activity against gastrointestinal cancer and breast cancer, and has entered the stage of clinical research, but its anticancer spectrum is narrow, and it has no effect on brain cancer. Budotitane, a metal anticancer drug developed in the 1980s, has good anticancer activity for solid tumors such as colorectal cancer.

#### 2.5.2. Other Non-Metallic Anticancer Drugs

Selenium (Se) is a trace element required by the human body and is involved in many biological reactions. Selenium has a balanced function for the thyroid, muscles, prostate, brain, and testes. Both organic and inorganic chemical forms are unique to natural selenium. Selenium intake is known to have a narrow range between inadequate and toxic levels [186]. It is therefore important to carefully monitor intake of this element. It is important to note that the recommended daily intake of selenium varies according to geographical area [187]. Many studies on selenium have shown that selenium has an inhibitory effect on a wide range of malignant tumors, including lung, bladder, colon, liver, stomach, thyroid, and prostate cancers; however, this inhibitory effect tends to vary depending on the type of selenium, the dose, and the type of cancer. Its mechanisms of action are also varied, including regulation of the cell cycle and apoptosis, antioxidant properties of selenium-containing proteins, regulation of neovascularization, influence on extracellular matrix, inhibition of histone deacetylase, neutralization of carcinogenic toxicity, repair of DNA damage, and regulation of the immune system [188]. Although many mechanisms are not clearly understood, many medical workers have tried to inhibit the development of cancer by increasing the intake of selenium in patients, and have achieved varying degrees of efficacy. A summary of these cases found that when selenium levels in the blood of cancer patients were increased to 120–160 ng/mL, these patients were able to fight some cancers more effectively than patients with selenium deficiency. However, when selenium levels in the blood are too high, selenium intake has the risk of promoting tumor tissue growth. Therefore, the intake of selenium needs to be controlled within a certain range in order to achieve a more ideal therapeutic effect. Se may also be used to treat cancer when combined with other drugs, chemotherapy, and radiotherapy. Shuo et al. [189] reviewed the use of NK cell immunotherapy and pemetrexed (Pem) chemotherapy in the management of tumors. However, the overexpression of NK cell inhibitory receptors on the surface of tumor cells and the lower intracellular internalisation efficiency of Pem severely limited its clinical application. Therefore, they designed a set of selenium-based nanoparticles to synergistically improve proton pump chemotherapy and NK cell immunotherapy. The nanoparticles can release Pem to the tumor site via selenite produced by β-selenate oxidation and enhance the chemotherapeutic efficacy of Pem. In addition, selenium can block the production of NK cell inhibitory genes, thereby activating the immune capacity of NK cells. This strategy has a promising future in chemoimmunotherapy, as in vitro and in vivo experiments have revealed the possible mechanisms of chemoenhancement and immune activation by selenite. Liu et al. [190] used cell CIK-mediated immunotherapy as a typical adoptive cell transfer modality to treat cancer. However, the short-term persistence of CIK cells in the body and the complex tumor environment are major challenging factors for CIK-based immunotherapy. Therefore, they have shown a safe and efficient strategy for effective cancer immunotherapy by binding SeNPs to CIK cells. Interestingly, SeNPs can effectively prolong the retention time of CIK cells in the blood. In addition, SeNPs can substantially promote the cytotoxicity of CIK cells from cancer patients against tumor cells by increasing the expression of the activating receptor—NKG2D and its ligand, but are not toxic to CIK and tumor cells. A number of lines of evidence suggest that SeNPs are metabolized to selenocysteine, which is the main reason for their unique advantages in promoting CIK therapy. Importantly, this strategy can effectively induce natural killer cells to infiltrate the tumor and to polarize tumor-associated macrophages to the M1 phenotype, thereby triggering a potent immune response against the progression of a variety of tumors, including liver, breast, and prostate. Therefore, this study is a novel strategy for the advancement of clinical application of CIK therapy in the management of cancer.

## 3. Conclusions

In the treatment of cancer, metallic anticancer drugs play an important role. Structurally and in terms of research protocols, these metal drugs are very stable in nature and can be used as prodrugs against malignant tumors. They can be used alone or in combination with other drugs or chemicals to produce novel metallic anticancer drugs with improved efficacy. These novel anticancer drugs bring more hope to the clinical treatment of malignant tumors, offering patients more choices and better therapeutic effects. However, there is still a certain gap between the clinical performance of metal anticancer drugs and people’s expectations. At present, more in-depth studies are needed because the pathogenesis of cancer and the mechanism of action of certain metal complex anticancer drugs in vivo are still unclear. On the one hand, we need to conduct in-depth studies on the pathogenesis of tumor diseases at the cellular level to establish the relationship between cancer cell growth and certain intracellular components.

The most important problem to be tackled today is that the principle of action and the process of altering the biological functions of metal-based anticancer drugs and cells are particularly important, and the course of the reactions can be studied in terms of kinetics and thermodynamics. The process of drug synthesis involves many complex organic reactions and special reaction conditions and other factors, and because there are no specific tools to keep these conditions constant, various kinds of side reactions and by-products occur. Whether these non-primary substances will be harmful to the human body, and how their presence in the human body will further affect the anticancer drugs, has yet to be unearthed and thoroughly investigated by researchers. In biological experiments and clinical applications, it is also a significant problem that volunteers cannot fully express in technical terms the side reactions and effects that occur when the drugs enter the body. Due to the constraints of ethical and moral norms, some side effects and effects on various organs of the human body cannot be collected first-hand, so experimental data collection is also an urgent issue. Although today’s high-tech instruments play an inestimable role in assisting research, biological experiments still lack a certain theoretical basis, which does not allow us to see the reactions at the root of the drugs and at the level of biomolecules. An organism is an individual that undergoes organic reactions at all times and in all places, and further research into anticancer drugs and biomolecules is crucial not only for the treatment of cancer, but also for the further study of the evolution of organisms and the slowing of the aging process. With the further understanding of the anticancer mechanism of metal complexes and their conformational relationship, the anticancer prospect of metal complexes will be broader.

## Figures and Tables

**Figure 1 pharmaceutics-15-02750-f001:**
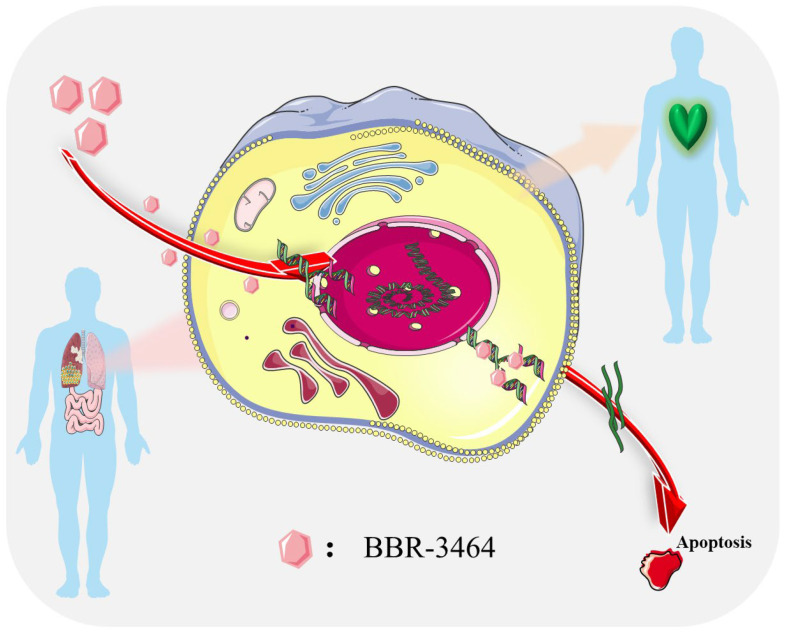
Schematic diagram of the anticancer mechanism of metal anticancer drug BBR-3464.

**Figure 2 pharmaceutics-15-02750-f002:**
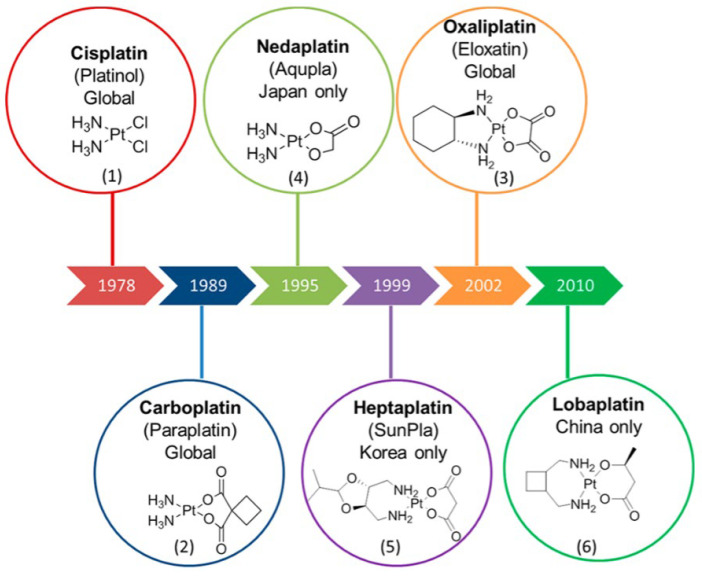
Development of classical platinum anticancer drugs [14].

**Figure 3 pharmaceutics-15-02750-f003:**
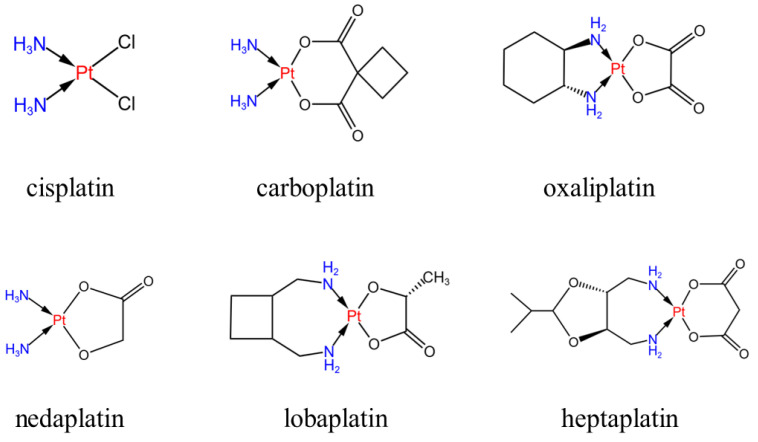
Classical platinum anticancer drugs.

**Figure 4 pharmaceutics-15-02750-f004:**
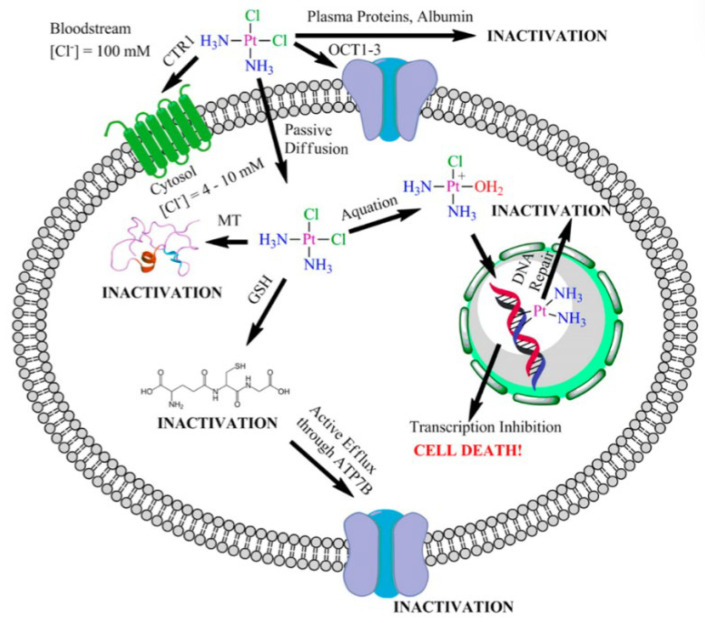
Anticancer mechanism of cisplatin [1].

**Figure 5 pharmaceutics-15-02750-f005:**
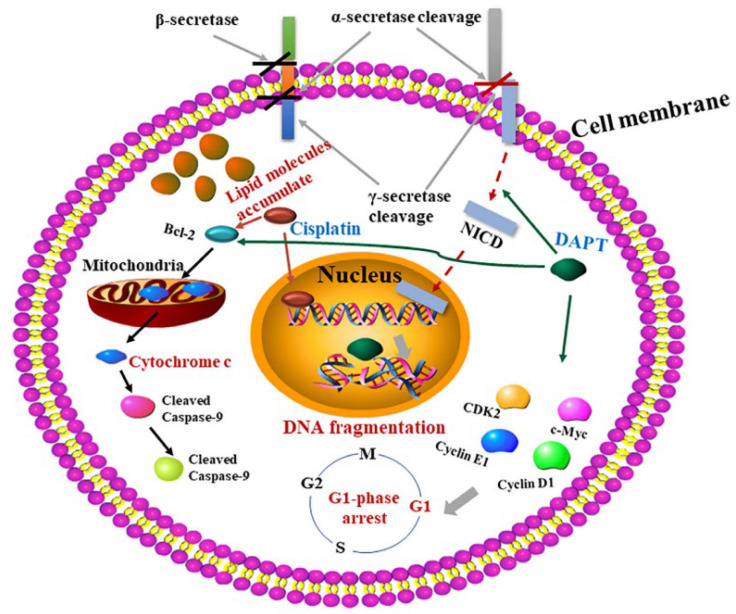
Schematic diagram of DAPT combined with cisplatin in the treatment of osteosarcoma cells [33].

**Figure 6 pharmaceutics-15-02750-f006:**
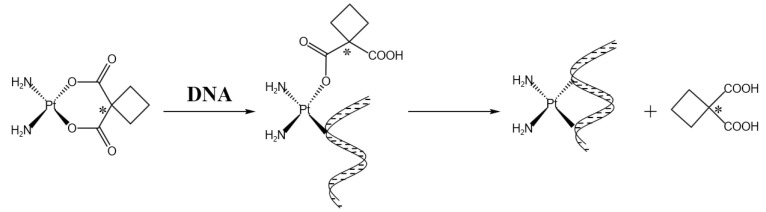
Metabolic steps of carboplatin. “*” stands for ^14^C.

**Figure 7 pharmaceutics-15-02750-f007:**
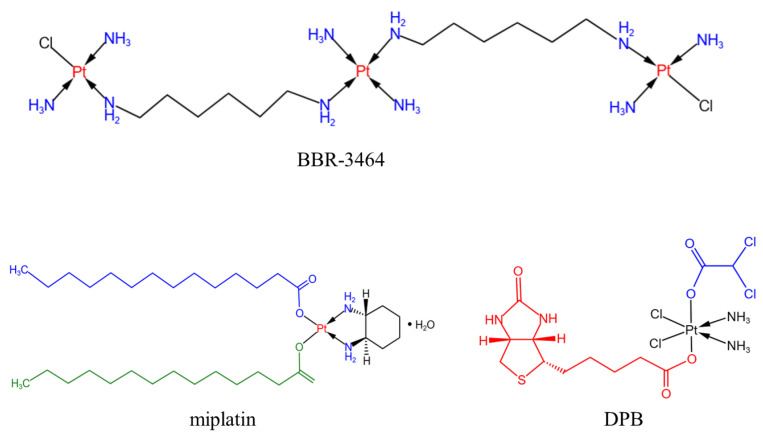
Non-classical platinum anticancer drugs.

**Figure 8 pharmaceutics-15-02750-f008:**
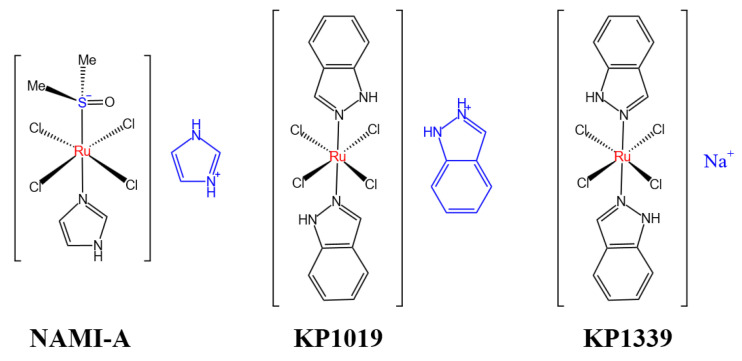
Chemical structure of NAMI-A, KP1019, and KP1339.

**Figure 9 pharmaceutics-15-02750-f009:**
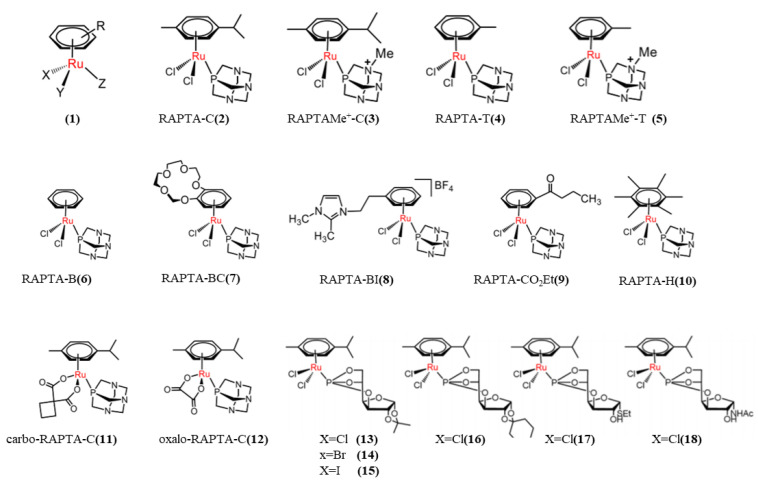
**1** is the general structural formula of the semi-filled arylruthenium (II) complex; **2**~**12** are the chemical structures of RAPTA compounds. **13**~**18** are the molecular structure formula of arylruthenium (II) containing glucose molecules [112].

**Figure 10 pharmaceutics-15-02750-f010:**
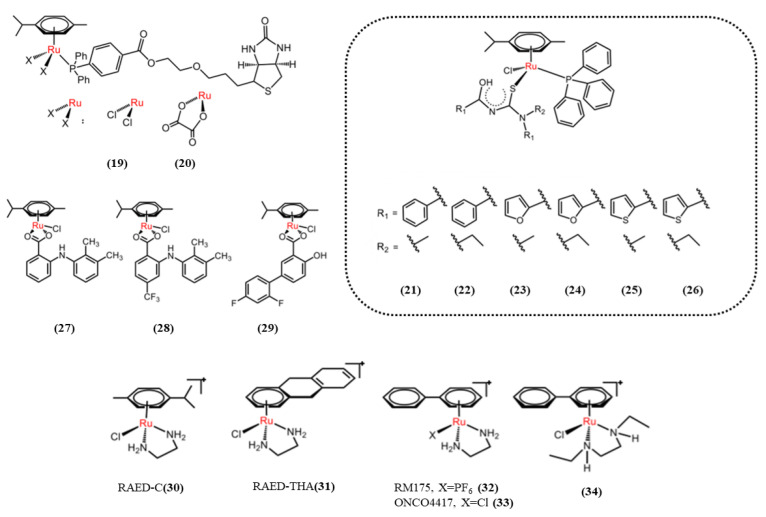
**19** and **20** are arylruthenium (II) complexes of biotin-functionalized monodentate phosphorus ligands; **21**~**26** are monodontic acylthiourea aryl ruthenium (II) complexes; **27**~**29** are the arylruthenium (II) complexes of SAID class. **30**~**34** are structural formulas for ethylenediamine arylruthenium (II) complexes [112].

**Figure 11 pharmaceutics-15-02750-f011:**
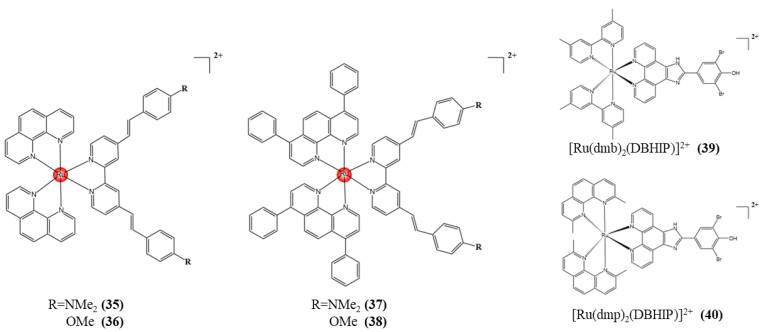
Structure diagram of ring metal Ruthenium complex **35**~**38** and [Ru(dmb)_2_(DBHIP)]^2+^ [129,130].

**Figure 12 pharmaceutics-15-02750-f012:**
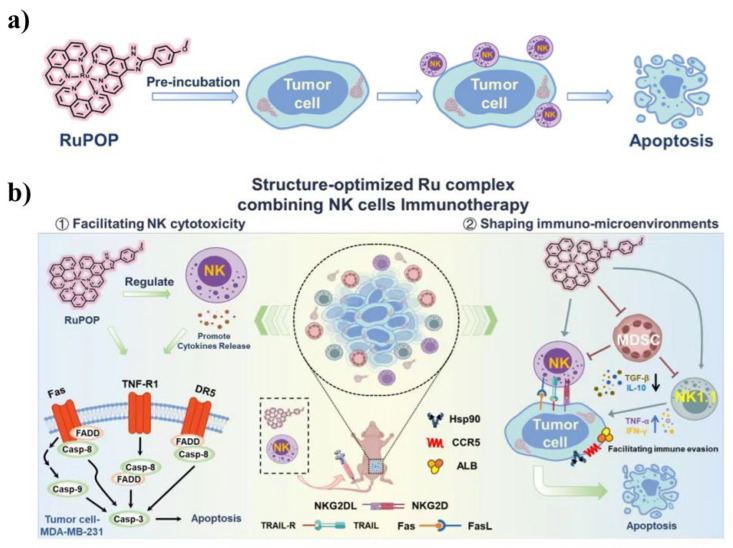
(**a**) RuPOP combined with hNKs to promote the killing of MDA-MB-231 tumor cells; (**b**) Structurally optimized RuPOP complexes promote NK cell immunotherapy for triple-negative breast cancer [131].

**Figure 14 pharmaceutics-15-02750-f014:**
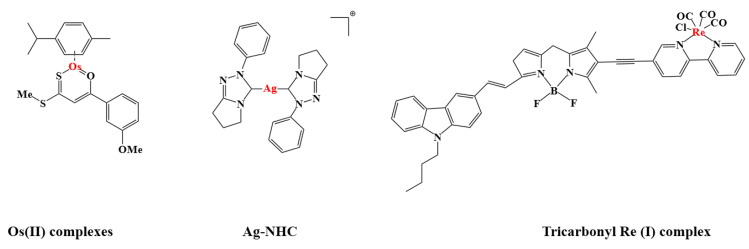
Structural schematic diagram of osmium, silver, and rhenium complexes.

## Data Availability

All data and materials in this study are included in the published article.

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
