# Peer review of "Research Progress of Metal Anticancer Drugs"

_pharmaceutics, 2023, doi:10.3390/pharmaceutics15122750_

Round 1

Reviewer 1 Report

Comments and Suggestions for Authors

The manuscript by Y. Bai et al reviews the recent scientific literature on Pt, Ru, Ir, Au and other anticancer metal complexes. Although there are some recent reviews on the use of transition metal complexes as anticancer drugs (such as B. Englinger et al. “Metal drugs and the anticancer immune response”, Chem Rev. 2019, 119, 1519-1624, not cited in this manuscript), there are some more recent examples here. In my opinion the manuscript can be accepted for publication in pharmaceutics after some issues have been addressed.

- I had to search for the meaning of qi, which is cited in page 4 a couple of times (“supplementing qi an blood…” “with positive qi weakness…”). It seems that qi refers to “the vital energy of the body in Traditional Chinese medicines (TCM), the flow of which must be unimpeded for health”. Qi is thus a pseudoscientific and unverified concept, and I think that a scientific journal such as Pharmaceutics is not the appropriate medium to discuss about these concepts. Please, remove these comments and refer only to scientifically verified facts.

- Add the reference B. Englinger et al. “Metal drugs and the anticancer immune response”, Chem Rev. 2019, 119, 1519-1624

- Figure 1 must be improved (the chemical structure is not clear) and must by cited in the text.

- In page 11, what is the structure of KP1339?

- The title of section 2.2.1, change “arylruthenium” by “ruthenium(II) arene” or “ruthenium(II) arene complexes”.

- Change the title of section 2.2.2, “Ring metal ruthenium” by “Ruthenium(II) polypyridine complexes”

- Add a new Figure with some of the iridium and gold complexes mentioned in the review.

- In section 2.3.2 of page 19, why again the same title “Iridium anticancer drugs” as in section 2.3.? Which is the difference with section 2.3.1? Also, in section 2.3.2., “Iridium cyclometallide” and “cyclometallized iridium” should be in both cases “cyclometalated iridium(III)”.

- Add a new Figure with the structure of some of the mentioned metal drugs cited in section 2.5.1.

- In section 2.5.2 “Other non-metallic anticancer drugs”, there are many other non-metallic anticancer drugs. Why only selenium containing drugs is commented here?

- In some of the references, the initial and final pages are missing

- Some other minor mistakes: Figure 4 mentioned in page 9 should be Figure 5; in the same page “loplatin” would probably be “lubaplatin”; Figure 1.3 mentioned in page 11 doesn’t appear in the manuscript.

- Finally, the English in some parts of the text must be improved for better understanding.

Author Response

The manuscript by Y. Bai et al reviews the recent scientific literature on Pt, Ru, Ir, Au and other anticancer metal complexes. Although there are some recent reviews on the use of transition metal complexes as anticancer drugs (such as B. Englinger et al. “Metal drugs and the anticancer immune response”, Chem Rev. 2019, 119, 1519-1624, not cited in this manuscript), there are some more recent examples here. In my opinion the manuscript can be accepted for publication in pharmaceutics after some issues have been addressed.

- I had to search for the meaning of qi, which is cited in page 4 a couple of times (“supplementing qi an blood…” “with positive qi weakness…”). It seems that qi refers to “the vital energy of the body in Traditional Chinese medicines (TCM), the flow of which must be unimpeded for health”. Qi is thus a pseudoscientific and unverified concept, and I think that a scientific journal such as Pharmaceutics is not the appropriate medium to discuss about these concepts. Please, remove these comments and refer only to scientifically verified facts.

Our reply: Based on your valuable suggestions, we have deleted the part about "Qi", and the revised part is as follows:

Natural terpenoids also have strong anti-tumor activity [26], and play a synergistic anti-breast cancer effect when combined with cisplatin. Other natural active ingredients such as polysaccharides have also been combined with cisplatin for anti-breast cancer research [27]. Shenqi Fuzheng injection can down-regulate the expression of P-glycoprotein in MDA-MB-231 cells resistant to cisplatin, ……

- Add the reference B. Englinger et al. “Metal drugs and the anticancer immune response”,Chem Rev. 2019, 119, 1519-1624

Reply: Based on your valuable suggestions, we have cited this literature and added the appropriate information in the revised version. The modifications are as follows:

Besides their extensively researched cytotoxic and anticancer properties, metal-based drugs are commonly employed in cancer-immune interactions and have the ability to reverse pivotal aspects of immune evasion [11].
[11]Bernhard E, Christine P, Petra H, et al. Metal Drugs and the Anticancer Immune Response[J]. Chemical reviews, 2019, 119(2): 1519-1624.

- Figure 1 must be improved (the chemical structure is not clear) and must by cited in the text.

ReplyBased on your valuable suggestions, we have improved and optimised the chemical structure in Figure 1, which is quoted in the text, and added relevant information. The chemical structure of BBR-3464 is shown in Figure 7. The modifications are as follows:

    Example, BBR-3464 (structure shown in Figure 7) can be used for the treatment of advanced cancer patients resistant to some common platinum anticancer drugs. The anticancer mechanism is shown in Figure 1, which has a higher degree of cross-linking with DNA compared to cisplatin, powerful anticancer efficacy and no crossresistance.

Figure 1. Schematic diagram of the anticancer mechanism of metal anticancer drug BBR-3464.

Figure 7.Non-classical platinum anticancer drugs

- In page 11, what is the structure of KP1339?

Reply: Thanks to your constructive comments, we have drawn the chemical structure of KP1339 in the revised manuscript as shown in Figure 8.

Figure 8. Chemical structure of NAMI-A、KP1019 and KP1339.

-The title of section 2.2.1, change “arylruthenium” by “ruthenium(II) arene” or “ruthenium(II) arene complexes”.

Reply: Based on your valuable suggestion, we have changed the title to "Ruthenium(II) arene complexes". The change is as follows:

2.2.1 Ruthenium(II) arene complexes

- Change the title of section 2.2.2, “Ring metal ruthenium” by “Ruthenium(II) polypyridine complexes”

ReplyBased on your valuable suggestion, we have changed the title to "Ruthenium(II) polypyridine complexes". The modification is as follows:

2.2.2 Ruthenium(II) polypyridine complexes

- Add a new Figure with some of the iridium and gold complexes mentioned in the review.

ReplyThanks to your constructive comments, we have drawn some of the complexes mentioned in the text about iridium and gold in the revised version. As shown in Fig. 13:

Figure 13. Structural schematic diagram of iridium and gold complexes.

- In section 2.3.2 of page 19, why again the same title “Iridium anticancer drugs” as in section 2.3.? Which is the difference with section 2.3.1? Also, in section 2.3.2., “Iridium cyclometallide” and “cyclometallized iridium” should be in both cases “cyclometalated iridium(III)”.

ReplyThank you very much for your delicate review of the comments made, due to our carelessness the title of 2.3.2 was written incorrectly. Our title in the revised version is: 2.3.2 Cyclometallic iridium anticancer drugs, the difference between 2.3.1 and 2.3.2 will be added in the revised version with relevant information. The revision is as follows: Because iridium-based anticancer drugs are iridium-containing metal complexes, they can be divided into the following two main categories due to the difference in the groups connected to iridium: aryl complexes and cyclic metal complexes. Therefore, they are divided into aryl iridium anticancer drugs and cyclometalated iridium(III) anticancer drugs for presentation in the following. We have changed "Iridium cyclometallide" and "cyclometallised iridium" to "Cyclometalated iridium(III)" in the revised version and have corrected them in full below.

- Add a new Figure with the structure of some of the mentioned metal drugs cited in section 2.5.1.

ReplyBased on your valuable suggestions, we have included some chemical structure diagrams of metal drugs in section 2.5.1, as shown in Figure 14.

Figure 14. Structural schematic diagram of osmium, silver, and rhenium complexes.

- In section 2.5.2 “Other non-metallic anticancer drugs”, there are many other non-metallic anticancer drugs. Why only selenium containing drugs is commented here?

ReplyBecause of the large number of experimental papers and review papers read on the way to writing this manuscript, it was discovered that out of the metallic anticancer drugs, selenium anticancer drugs are particularly prominent among the non-metallic anticancer drugs. Selenium is necessary for the well-balanced function of many organs such as the thyroid, brain, muscles, prostate, and testes. Many studies on selenium have shown that selenium has a certain inhibitory effect on a variety of cancers, including lung, bladder, colorectal, liver, stomach, thyroid, prostate, etc. However, this inhibitory effect often varies depending on the species of selenium, the dose, and the type of cancer. The mechanisms of action also vary, including regulation of the cell cycle and apoptosis, antioxidant properties of selenium-containing proteins, regulation of neovascularisation, effects on the extracellular matrix, inhibition of histone deacetylase, neutralisation of carcinogen toxicity, repair of DNA damage, and regulation of the immune system [177].

- In some of the references, the initial and final pages are missing.

ReplyThank you for your valuable comments, we have added the corresponding start and end page numbers in the revised version.

- Some other minor mistakes: Figure 4 mentioned in page 9 should be Figure 5; in the same page “loplatin” would probably be “lubaplatin”; Figure 1.3 mentioned in page 11 doesn’t appear in the manuscript.

ReplyThanks to your careful review, we have corrected and modified all the wrong versions due to personal errors. The text and images in the revised version are compatible and optimised. The typo "loplation" has been changed to "lubaplation".

- Finally, the English in some parts of the text must be improved for better understanding.

ReplyThank you for your valuable comments, we have thoroughly checked and revised the manuscript. Inappropriate grammatical knowledge and inappropriate adjectives and positions have been replaced and rewritten.

Reviewer 2 Report

Comments and Suggestions for Authors

The authors submitted a review article summarizing the recent "Research Progress of Metal Anticancer Drugs". My main criticism is that too many older references were cited, but too little attention was paid to the latest work. The authors should please carry out another literature search. In addition, the article urgently needs to be revised with regard to spelling mistakes, typos, inaccuracies and style. Above all, the bibliography must also be carefully improved with regard to author names (order of family name/first name), page numbers, etc. 

The following small selection of inaccuracies should be noted:

page 4: "alsine" to "alanine"; "Zoledronic" to "zoledronic"

page 6: The authors are requested to explain the "six-membered ring with DNA" more in detail (formular?) ref [33] is unclear.

page 7: “cise" to "cis"

page 8: I would prefer "(1R,2R)-" instead of L-

page 10: Regarding Pt(IV) complexes, D. Gibson´s work was completely

ignored. Please check these papers.

page 12: ref [87] is wrong; who is "Tocher" from 1992?

page 19: "quadrangular planar quadrilateral" change to "square-planar"

page 21: following recent reference describing cytotoxic Os complexes should be added: Int. J. Mol. Sci. 2022, 23(9), 4976

Comments on the Quality of English Language

The language and references need to be thoroughly polished

Author Response

The authors submitted a review article summarizing the recent "Research Progress of Metal Anticancer Drugs". My main criticism is that too many older references were cited, but too little attention was paid to the latest work. The authors should please carry out another literature search. In addition, the article urgently needs to be revised with regard to spelling mistakes, typos, inaccuracies and style. Above all, the bibliography must also be carefully improved with regard to author names (order of family name/first name), page numbers, etc. The following small selection of inaccuracies should be noted:

page 4: "alsine" to "alanine"; "Zoledronic" to "zoledronic"

Our replyThank you for your valuable comments, we have corrected the typos in the revised version.

    Xin et al. studied the efficacy of alanine(审稿人2-1)-proline-arginine-proline-glycine (APRPG) peptide-coupled polyglycol cationic liposome coated with zoledronic acid (ZOL) (APRPG-PEG-ZOL-CLPS) for vascular normalization[32].

page 6: The authors are requested to explain the "six-membered ring with DNA" more in detail (formular?) ref [33] is unclear.

Our replyThank you for your valuable comments.

    Carboplatin forms a six-membered ring with DNA, showing greater stability and water solubility [39]. The metabolism of carboplatin requires two steps (as shown in Figure 5), one of which is the opening of the six-membered ring, a slower but critical reaction process. The second is the dissociation of ligands, which is relatively fast [40].

Figure 5. Metabolic steps of carboplatin.

page 7: “cise" to "cis"

Our replyThank you for your valuable comments, we have corrected the typos in the revised version. Nedaplatin, also known as cis-glycolate diammine platinum (as shown in Figure 6)……

page 8: I would prefer "(1R,2R)-" instead of L

Our replyThank you for your valuable comments, we have replaced them.

Oxaliplatin's chemical name is (1R,2R)-trans-diamine cyclohexane oxalate platin, also known as oxalate platin, which belongs to the third generation of platinum anti-cancer drugs.

page 10: Regarding Pt(IV) complexes, D. Gibson´s work was completely ignored. Please check these papers.

Our replyThanks to your constructive comments, we have referenced and cited the work of D. Gibson. Below is the added information and reference content.

  1. Gibson et al.[34] reported on the combination of cisplatin and caffeic acid for the treatment of cancer cells. The results show that caffeic acid is a dual acting drug that can sensitise or habituate cells to cisplatin therapy depending on the time of administration. The co-administration of caffeic acid and cisplatin was found to be effective, which provided a reasonable basis for the preparation of a new platinum caffeic acid compound.

[34]Sirota R, Gibson D, Kohen R. The timing of caffeic acid treatment with cisplatin determines sensitization or resistance of ovarian carcinoma cell lines[J]. Redox Biology, 2017, 11: 170-175.

page 12: ref [87] is wrong; who is "Tocher" from 1992?

Our replyThank you for your valuable comments and the inconvenience caused due to our negligence. We have cited the correct literature and corrected it in the revised version.

  page 19: "quadrangular planar quadrilateral" change to "square-planar"

Our replyThank you for your valuable comments, we have replaced the inappropriate words in the manuscript.

    Au3+ has the same d8 closed shell electronic structure as Pt2+, and can form a square-planar structure.

page 21: following recent reference describing cytotoxic Os complexes should be added: Int. J. Mol. Sci. 2022, 23(9), 4976.

    Jana Hildebrandt et al.[175] reported on various ruthenium (II) complexes and promising osmium (II) analogues, all of which use cinnamic acid as an O, S bidentate ligand. And the anticancer activity and ability to evade platinum resistance mechanisms of these compounds were studied. The results showed that Ru (II) and Os (II) compounds (as shown in Figure 14)containing O and S bidentate ligands have high cytotoxicity, but their effects on DNA damage and cell cycle are not strong. This may be the basis for bypassing drug resistance mechanisms and high specificity of cancer cells.

[175]Jana H, Norman H, Daniel K, et al. Highly Cytotoxic Osmium(II) Compounds and Their Ruthenium(II) Analogues Targeting Ovarian Carcinoma Cell Lines and Evading Cisplatin Resistance Mechanisms[J]. International Journal of Molecular Sciences, 2022, 23(9): 4976.

Comments on the Quality of English Language the language and references need to be thoroughly polished.

Our replyThank you for your valuable comments, we have thoroughly checked and revised the manuscript. Inappropriate grammatical knowledge and inappropriate adjectives and positions have been replaced and rewritten.

Reviewer 3 Report

Comments and Suggestions for Authors

This study provides a concise overview of the topic of metal anticancer drugs and their potential as therapeutic agents. It effectively highlights the limitations of traditional chemotherapy and the need for new drug designs and treatment strategies.

I have provided some suggestions that might improve the quality of the paper below;

1.      Most of the figures used in this review paper have a low resolution. The authors should provide higher resolution figures.

2.     Expand the review paper with recent research/review papers references. There are only – about– 25 references from 2023-2018 out of 183 Ref. I have provided you some suggestions to be reffrences below;

·      https://doi.org/10.1002/asia.202200270

·      https://doi.org/10.1155/2022/6493458

·      doi: 10.3390/molecules27196485

·      doi: 10.1042/BSR20212160

·      doi: 10.1039/d0cs01075h.

·      DOI: 10.2174/1381612827666210916140627

·      doi: 10.1039/d3dt00366c.

·      doi: 10.2147/DDDT.S275007.

3.      There are a few instances where information could be expanded upon or clarified. For example, when discussing the clinical effects of oxaliplatin, it would be helpful to provide specific examples or references to support the claims made.

4.     Consider providing more context and background information for the different drugs. For instance, you mention that oxaliplatin belongs to the third generation of platinum anti-cancer drugs, but it would be beneficial to explain what distinguishes the different generations and why it is relevant.

5.     It would be beneficial to provide a brief summary or key results of each study to give readers a better understanding of the experimental findings.

6.     In the conclusion, it would be helpful to include any limitations or challenges associated with the use of metal-based anticancer drugs.

Comments on the Quality of English Language

- The authors should proofread the text for grammar and clarity.

- Some sentences are lengthy and complex, making them difficult to comprehend. Try breaking them down into shorter, more concise sentences to improve readability.

Author Response

This study provides a concise overview of the topic of metal anticancer drugs and their potential as therapeutic agents. It effectively highlights the limitations of traditional chemotherapy and the need for new drug designs and treatment strategies. I have provided some suggestions that might improve the quality of the paper below;

  1. Most of the figures used in this review paper have a low resolution. The authors should provide higher resolution figures.

Our replyThank you for your valuable comments, we have optimised most of the images. They are shown below:

Figure 2. Development of classical platinum anticancer drugs. [193]

Figure 3.Anticancer mechanism of cisplatin [194]

Figure 4. Schematic diagram of DAPT combined with cisplatin in the treatment of osteosarcoma cells [33]

Figure 6. Classical platinum anticancer drugs

Figure 7.Non-classical platinum anticancer drugs

Figure 8. Chemical structure of NAMI-A、KP1019 and KP1339

Figure 11. Structure diagram of ring metal Ruthenium complex 35~38 and [Ru(dmb)2(DBHIP)]2+ [130,196]

  1. Expand the review paper with recent research/review papers references. There are only – about– 25 references from 2023-2018 out of 183 Ref. I have provided you some suggestions to be reffrences below;
  • https://doi.org/10.1002/asia.202200270
  • https://doi.org/10.1155/2022/6493458
  • doi: 10.3390/molecules27196485
  • doi: 10.1042/BSR20212160
  • doi: 10.1039/d0cs01075h.
  • DOI: 10.2174/1381612827666210916140627
  • doi: 10.1039/d3dt00366c.
  • doi: 10.2147/DDDT.S275007.

Our replyThank you for your constructive comments, we have carefully read the literature you provided and cited it in the corresponding places in the text. The modifications are as follows:

At present, the methods of treating cancer include targeted anti-cancer drug therapy, chemotherapy, hormone treatment, radiotherapy, surgery and so on. These treatments are selected and combined according to the patient's specific condition and tumor type to achieve the best treatment results [3,4]. Nanotechnology has enormous potential in the prognosis, diagnosis, and drug delivery of cancer, and is therefore considered to have significant applications in clinical treatment [5].

Platinum-based drugs, especially cisplatin, the first generation of platinum-based anticancer drug, are the most commonly used anticancer drugs in clinical practice, and its therapeutic effect on head and neck cancer and reproductive system cancer is accurate[12].

For example, research mixed Gd-Pt therapeutic agents that incorporate platinum into micelles or other types of nanoparticles [31].

NAMI-A has little activity against primary tumors, but is highly active against secondary tumors such as non-small cell lung cancer[88].

Jana Hildebrandt et al.[175] reported on various ruthenium (II) complexes and promising osmium (II) analogues, all of which use cinnamic acid as an O, S bidentate ligand. And the anticancer activity and ability to evade platinum resistance mechanisms of these compounds were studied. The results showed that Ru (II) and Os (II) compounds (as shown in Figure 14)containing O and S bidentate ligands have high cytotoxicity, but their effects on DNA damage and cell cycle are not strong. This may be the basis for bypassing drug resistance mechanisms and high specificity of cancer cells[176].

Hessam Hosseinkazemi et al. [184] reported on the potential use of iron oxide nanoparticles as therapeutic drugs and acknowledged their superiority over traditional therapeutic drugs. Although only used for the treatment of iron deficiency anemia and cancer, multiple experiments are currently being conducted to promote its clinical application.

In addition, copper complexes can also cause ROS to accumulate in large quantities in cells, leading to apoptosis [186].

[4]Liana R LCorina A H, Bogdan S, et al. Metallo-Drugs in Cancer Therapy: Past, Present and Future[J]. Molecules, 2022, 27(19): 6485.

[5]Ahmed F, Ahmad J, Khan A M, et al. Recent Advances in Theranostic Applications of Nanomaterials in Cancer[J]. Current Pharmaceutical Design, 2022, 28(2):133-150.

[12]P S V , Shubhankar G, Ravi K T, et al. Challenges and opportunities in the development of metal-based anticancer theranostic agents[J]. Bioscience reports, 2022, 42: 1-25.

[31] Robertson G A, Rendina M L. Gadolinium theranostics for the diagnosis and treatment of cancer [J]. Chemical Society reviews, 2021,50(7): 4231-4244.

[88]Yeul S L, Young C K, TaeGyu N. Ruthenium Complexes as Anticancer Agents: A Brief History and Perspectives [J]. Drug design, development and therapy, 2020,14: 5375-5392.

[175]Jana H, Norman H, Daniel K, et al. Highly Cytotoxic Osmium(II) Compounds and Their Ruthenium(II) Analogues Targeting Ovarian Carcinoma Cell Lines and Evading Cisplatin Resistance Mechanisms[J]. International Journal of Molecular Sciences, 2022, 23(9): 4976.

[184]Hessam H, Saeed S, Andrew O, et al. Applications of Iron Oxide Nanoparticles against Breast Cancer[J]. Journal of Nanomaterials, 2022, 2022: 6493458.

[186] Njenga W L, Mbugua N S, Odhiambo A R, et al. Addressing the gaps in homeostatic mechanisms of copper and copper dithiocarbamate complexes in cancer therapy: a shift from classical platinum-drug mechanisms[J]. Dalton transactions, 2023, 52: 5823-5847.

  1. There are a few instances where information could be expanded upon or clarified. For example, when discussing the clinical effects of oxaliplatin, it would be helpful to provide specific examples or references to support the claims made.

Our replyThank you for your valuable comments, we have mentioned in the main text section about the clinical use of oxaliplatin as detailed below:

Cui [65] explores the clinical efficacy of oxaliplatin combined with chemotherapy in the treatment of colon cancer. Oxaliplatin combined with chemotherapy has a significant effect on colon cancer patients. Significantly increase TNF-αand IL-2 levels can effectively improve clinical symptoms and improve patients' quality of life. Lan et al. [66] studied the efficacy of oxaliplatin combined with tigio in the treatment of advanced esophageal cancer and its impact on patient survival time. The results showed that oxaliplatin combined with Tigor in the treatment of advanced esophageal cancer patients can improve the level of tumor markers, improve drug safety and prolong survival time.

  1. Consider providing more context and background information for the different drugs. For instance, you mention that oxaliplatin belongs to the third generation of platinum anti-cancer drugs, but it would be beneficial to explain what distinguishes the different generations and why it is relevant.

Our replyThank you for your valuable comments, we have mentioned in the main text section about the benefits and improvements between the different drugs. The details are given below:

The structure is similar to that of cisplatin, where the two NH3's of cisplatin remain unchanged and the two Cl's are replaced by the chelating coordination of dicarboxylic acid groups. Compared with cisplatin, the clinical therapeutic effect is more significant, the adverse reactions are less severe, and it is soluble in water, so it is very convenient to use [35].

Although it is a platinum metal drug like cisplatin and carboplatin, the most significant thing is that it does not produce cross-resistance, so the clinical effect is very good when treating carboplatin and cisplatin resistant malignant tumors.

  1. It would be beneficial to provide a brief summary or key results of each study to give readers a better understanding of the experimental findings.

Our replyThanks to your valuable comments, we have made a table based on the main types of anticancer drugs, principles of reaction and areas of application and important results as a way to make it easier for the reader to read and understand. The table is shown below:

Table 1. Types of anticancer drugs, reaction principles and outcomes.

  1. In the conclusion, it would be helpful to include any limitations or challenges associated with the use of metal-based anticancer drugs.

Our replyThank you for your constructive comments and we have rewritten the conclusion section. The details are as follows:

The most important problem to be tackled today is that the principle of action and the process of altering the biological functions of metal-based anticancer drugs and cells are particularly important, and the course of the reactions can be studied in terms of kinetics and thermodynamics. The process of drug synthesis involves many complex organic reactions and special reaction conditions and other factors, and because there are no specific tools to keep these conditions constant, various kinds of side reactions and by-products occur. Whether these non-primary substances will be harmful to the human body, and how their presence in the human body will further affect the anti-cancer drugs, has yet to be unearthed and thoroughly investigated by researchers. In biological experiments and clinical applications, it is also a significant problem that volunteers cannot fully express in technical terms the side reactions and effects that occur when the drugs enter the body. Due to the constraints of ethical and moral norms, some side effects and effects on various organs of the human body cannot be collected first-hand experimental data is also an urgent issue. Although today's high-tech instruments play an inestimable role in assisting research, biological experiments still lack a certain theoretical basis, which does not allow us to see the reactions at the root of the drugs and at the level of biomolecules. An organism is an individual that undergoes organic reactions at all times and in all places, and further research into anti-cancer drugs and biomolecules is crucial not only for the treatment of cancer, but also for the further study of the evolution of organisms and the slowing of the ageing process. With the further understanding of the anticancer mechanism of metal complexes and their conformational relationship, the anticancer prospect of metal complexes will be broader.

Comments on the Quality of English Language

- The authors should proofread the text for grammar and clarity.

Our replyThank you for your valuable comments, we have thoroughly checked and revised the manuscript. Inappropriate grammatical knowledge and inappropriate adjectives and positions have been replaced and rewritten.

- Some sentences are lengthy and complex, making them difficult to comprehend. Try breaking them down into shorter, more concise sentences to improve readability.

Our replyThank you for your valuable comments, we have rectified the complex statements in the text and broken down the long sentences that were difficult to understand into shorter ones.

Round 2

Reviewer 1 Report

Comments and Suggestions for Authors

The authors have satisfactorily addressed all suggested changes and the manuscript can be accepted as is.

Reviewer 2 Report

Comments and Suggestions for Authors

The manuscript can be accepted in its present form